

# Numerical simulations of windblown dust over complex terrain: The Fiambalá Basin episode in June 2015

Leonardo A. Mingari[1,6,7], Estela A. Collini[2,6], Arnau Folch[3], Walter Báez[4], Emilce Bustos[4], María Soledad Osores[1,5,6], Florencia Reckziegel[4], Peter Alexander[1,7], and José G. Viramonte[4]

[1]CONICET, Argentina
[2]Servicio de Hidrografía Naval (SHN), Argentina
[3]Barcelona Supercomputing Center (BSC), Barcelona, España
[4]INENCO-GEONORTE (UNSa–CONICET), Salta, Argentina
[5]Comisión Nacional de Actividades Espaciales (CONAE), Argentina
[6]Servicio Meteorológico Nacional (SMN), Argentina
[7]Instituto de Física de Buenos Aires (IFIBA), Argentina

*Correspondence to:* Leonardo A. Mingari (lmingari@gmail.com)

**Abstract.** On the 13 June 2015, the London Volcanic Ash Advisory Centre (VAAC) warned the Buenos Aires VAAC about a possible volcanic eruption from the *Nevados Ojos del Salado* volcano (6,879 m), located in the Andes mountain range on the border between Chile and Argentina. A volcanic ash cloud was detected by the SEVIRI instrument on board the Meteosat Second Generation (MSG) satellites from 14:00 UTC on 13 June. Further studies concluded that the phenomenon was caused

by remobilization of ancient pyroclastic deposits (circa 4.5 Ka Cerro Blanco eruption) from the *Bolsón de Fiambalá* (Fiambalá Basin) in northwestern Argentina.

In this paper, we provide the first comprehensive description of the dust episode through observations and numerical simulations. We have investigated the spatio-temporal distribution of aerosols and the emission process over complex terrain to gain insight into the key role played by the orography and the condition that triggered the long-range transport episode.

Numerical simulations of windblown dust were performed using the WRF-ARW/FALL3D modeling system with meteorological fields downscaled to a spatial resolution of 2 km in order to resolve the complex orography of the area. Results indicated that favourable conditions to generate dust uplifting occurred in northern Fiambalá Basin, where orographic effects caused strong surface winds. According to short-range numerical simulations, dust particles were confined to near-ground layers around the emission areas. On the other hand, dust aerosols were injected up to 5-6 km high in central and southern regions

of the Fiambalá Basin, where intense ascending airflows are driven by horizontal convergence.

Long-range transport numerical simulations were also performed to model dust cloud spreading over northern Argentina. Results of simulated vertical particle column mass were compared with the MSG-SEVIRI retrieval product. We tested two numerical schemes: with the default configuration of the FALL3D model, we found difficulties to simulate transport through orographic barriers, whereas an alternative configuration, using a numerical scheme to more accurately compute the horizontal

advection in abrupt terrains, substantially improved the model performance.





# 1 Introduction

On the 13 June 2015, the London Volcanic Ash Advisory Centre (VAAC) warned the Buenos Aires VAAC about a possible volcanic eruption from the *Nevados Ojos del Salado* volcano, the highest volcano in the world (6,879 m), located in the Andes mountain range on the border between Chile and Argentina. A volcanic ash cloud was detected by the infrared channels of the Spinning Enhanced Visible and Infrared Imager (SEVIRI) instrument on board the Meteosat Second Generation (MSG) satellites from 14:00 UTC (11:00 LT) on 13 June. In collaboration with a research group specialized in Central Andes volcanoes, Collini et al. (2015) concluded that the phenomenon was caused by remobilization of ancient pyroclastic deposits from the *Bolsón de Fiambalá* (Fiambalá Basin).

The Fiambalá Basin is an intermontane depression with a basin floor elevation of around 1650 m situated along the eastern flanks of the Andes Mountain Range in northwestern Argentina (Carrapa et al., 2006; McPherson, 2008) at approximately 27°45' S, 67°45' W. The dunes of Fiambalá Basin are mainly a product of the aeolian reworked pyroclastic materials originated from the Cerro Blanco strong volcanic eruption, one of the greatest eruptions of the Central Andes during the Holocene, which occurred circa 4500 years ago (Báez et al., 2015; Fernandez Turiel et al., 2015). Wind activity continued to mobilize this pyroclastic material until today, turning the Fiambalá Basin in a major dust source in northwestern Argentina.

Several theoretical and experimental studies focus on Saharan and Asian dust (Marticorena et al., 1997; Middleton and Goudie, 2001; Shao and Dong, 2006). However, there are fewer studies regarding dust activity in South America (e.g. Middleton, 1986; Kurgansky et al., 2011; Gaiero, 2007). Specifically, areas with persistent dust activity in Central- and North-Western Argentina have been identified as major dust sources of South America (Prospero et al., 2002). The forecast of dust episodes in these regions is hindered by the scarcity of meteorological stations and the presence of an extremely complex topography. As a consequence, a significant uncertainty exists in the identification of dust sources, emission rates, transport mechanisms and particle characteristics. Recently, recurrent events of volcanic ash mobilized by wind in Patagonia caused multiple impacts on the environment for several months, such as severe deterioration of air quality and airport disruptions (Wilson et al., 2013), as a consequence of the volcanic eruptions of the Cordón Caulle volcanic complex in 2011 (Collini et al., 2013) and the Calbuco volcano in 2015 (Romero et al., 2016; Reckziegel et al., 2016). These eruptions blanketed with volcanic ash a vast area of the Patagonia in Argentina. An outstanding summary of the 2011 Cordón Caulle eruption by Elissondo et al. (2016) describes in detail the areas and activities that suffered the impact of the volcanic ash plume and deposition from the eruption as well as events of volcanic ash resuspension. The authors mention several types of impacts, including road closure, cuts of electric and water supply, discontinuation of activities in urban areas, health problems, agricultural and livestock disruptions, vegetation damage and animal death. In distal areas (1 cm or less of tephra deposit), the impact included agricultural and livestock disruptions, interruption of transport systems (closure of airports and roads) and discontinuation of activities during days of tephra fall or resuspension.

At present, several global and regional dust forecast systems are available (Tegen and Fung, 1994; Nickovic et al., 2001; Benedetti et al., 2014). Particularly, studies focused on the performance of dispersion models simulating concentrations of windblown dust in South America are scarce (e.g. Johnson et al., 2010; Gaiero et al., 2013). For example, events of volcanic





ash resuspension in Patagonia were recently simulated using the WRF-ARW/FALL3D modeling system with promising results (Folch et al., 2014; Reckziegel et al., 2016). Dust models simulate the main processes of the dust cycle: emission, transport and deposition (Tegen, 2003). Customarily, parametrization of dust emission depends on both soil conditions and surface wind stress (Shao, 2001; Kok et al., 2012). Soils with readily erodible sediments of fine particles and low soil moisture content

are most sensitive to dust emission (Marticorena and Bergametti, 1995). Description of the dust emission processes involves phenomena related to a wide range of scales, such as micro-scale (e.g. boundary layer mixing) or mesoscale (e.g. flow over mountain terrain), which may not be resolved by dust models (Tegen and Schulz, 2014).

    Orography alters the surface flow and consequently the emission and transport of dust in many ways (Knippertz and Stuut, 2014). For example, wind tunnel measurements demonstrated the influence of complex terrain on dust mobilization (Xuan

and Robins, 1994). Recently, a strong dust event in the complex terrain of the Dead Sea valley was simulated with the model system COSMO-ART (Vogel et al., 2009) using a horizontal grid spacing of $\sim 3\,\mathrm{km}$, producing a reasonable spatio-temporal distribution of near-surface dust concentration, consistent with available measurements (Kishcha et al., 2016). In addition, Liu and Westphal (2001) showed that a better resolved topography leads to improved dust simulations over Asia. In this context, the performance of dust models in complex terrains is challenging because: (i) local topography plays a key role in the description

of the temporal and spatial distribution of aerosol particles, as low-level wind is closely related to geographic features and the nature of the underlying surface, and (ii) almost all major dust sources are in semi-arid and arid low-lying regions or in lands adjacent to strong topographical highs (Prospero et al., 2002).

    In this paper, numerical simulations of windblown dust were performed using the FALL3D volcanic ash dispersal model (Costa et al., 2006; Folch et al., 2009) driven by the WRF-ARW meteorological model (Skamarock et al., 2008). We assessed

the ability of this modeling strategy to properly represent the emission and transport of dust in the complex terrain environment of the Andes range using the episode of 13 June 2015 as a test case. The meteorological fields were downscaled to a spatial resolution of $2\,\mathrm{km}$ to resolve the orographic features of the Fiambalá Basin. Subsequently, short- and large-range simulations were performed and the results compared with the MSG-SEVIRI retrieval product.

    This work is framed in the context of efforts for the development of operational forecasting capabilities to predict the

occurrence of volcanic ash mobilization in the Andean volcanic region. The final product is intended to provide support to Volcanic Ash Advisory Centres (VAAC) and air quality agencies.

    This work is organized as follows: section 2 provides a description of the region in terms of the geological setting with a review of the current understanding of the local dust activity. Section 3 details the modeling strategy applied to the episode of 13 June 2015 and summarizes the parametrizations of the emission scheme. An overview of the dust outbreak with the

meteorological conditions associated is included in section 4. In section 5, we present the results of numerical simulations, addressing the impact of the complex orography on the simulations. The findings are discussed and conceptually integrated in section 6.



## 2  Background

### 2.1  Geological setting

The Andes is a north-south orogen located along the western margin of South America and is related to the subduction of the Farallón-Nazca and Antarctica Plates beneath the South American Plate. The main topographic feature in Central Andes is the high elevated Altiplano-Puna *plateau*. The southern edge of the Altiplano-Puna *plateau* coincides with the ENE-trending Ojos del Salado-San Buenaventura (OSSB) volcanic lineament  (Álvarez et al., 2015, and references therein). This lineament, transversal to the Andean strike, starts at the Chilean Central Valley and extends for almost 300 km into the back-arc region. The Ojos del Salado-San Buenaventura volcanic lineament is a prominent mountain range formed by stratovolcanoes including some of the highest volcanoes in the world (e.g. Ojos del Salado, Tres Cruces and Incahuasi volcanoes) as well as by dome complexes and collapse calderas (Baker et al., 1987; Mpodozis et al., 1996; Seggiaro et al., 2000; Montero Lopez et al., 2010a, b, c; Báez et al., 2015; Bustos et al., 2015). Since Neogene times, shallowing of the subducting slab and lithospheric delamination explains the eastward broadening of the arc magmatism along ENE normal and right lateral neotectonic structures to form the Ojos del Salado-San Buenaventura volcanic lineament (Kay and Coira, 2009). The Ojos del Salado-San Buenaventura volcanic lineament includes some volcanic centers considered active (last eruption <10 ka) but there are no reports of confirmed historical eruptions in the region.

The Cerro Blanco Volcanic Complex (CBVC) is located in the eastern edge of the Ojos del Salado-San Buenaventura volcanic lineament (Fig. 1) and was defined as a nested calderas system with domes and pyroclastic deposits associated (Seggiaro et al., 2000; Viramonte et al., 2004, 2005; Arnosio et al., 2005). All erupted products are rhyolitic/rhyodacitic in composition and Middle Pleistocene-Holocene in age (Viramonte et al., 2008; Arnosio et al., 2008; Montero Lopez et al., 2010a). The CBVC had at least two large-scale eruptions (Volcanic Explosivity Index $\geq$ 6) over the past 100,000 years (Báez et al., 2015). Particularly the caldera forming Cerro Blanco eruption (4200 BP) constitutes one of the greater Holocene volcanic events in Central Andes (Fernandez Turiel et al., 2015; Báez et al., 2015). During the paroxysmal phase of the Cerro Blanco eruption pyroclastic density currents that spread radially as far as 35 km from the vent were generated (15 km$^3$ minimum bulk volume of ignimbrites) (Báez et al., 2015). Also a huge amount of ash fall deposits that covered much of North-Western Argentina was produced (110 km$^3$ bulk volume) (Fernandez Turiel et al., 2015).

The Bolsón de Fiambalá (Fiambalá Basin) is an intermontane depression located 40 km southward of Cerro Blanco caldera, in a transitional region between the Southern Puna and the Northern Pampean Ranges. The Fiambalá Basin floor is 2,000 m below the average elevation of the Southern Puna. During the Cerro Blanco eruption, the Fiambalá Basin received a large volume of pyroclastic fallout deposits and a lesser extent of ignimbrite deposits. The unconsolidated nature of the deposits related to the Cerro Blanco eruption (Báez et al., 2015) allowed the rapid remobilization of pyroclastic material deposited in nearby areas to the Fiambalá Basin. Wind activity continued to mobilize this pyroclastic material until today turning the Fiambalá Basin in one of the major dust sources in North-Western Argentina.





## 2.2 Dust activity in the region

Major dust sources in South America (SA) are located along a north-south oriented band of arid and semi-arid land from the Argentine Patagonia to the coast of Peru (Gaiero et al., 2013). Three areas have been identified as persistent dust sources in SA (Prospero et al., 2002): (i) Patagonia ($\sim 50°$- $39°$ S), (ii) central-western Argentina ($36°$- $26°$ S), and (iii) the Puna/Altiplano

Plateau ($\sim 26°$- $15°$ S). According to model-predicted annual emission rates in SA (Johnson et al., 2010), Patagonia is expected to contribute over 95 % of mineral dust emissions, with the remaining emissions attributed to regions in Western Argentina.

The dust activity in Argentina shows seasonal and diurnal patterns. For example, Middleton (1986) mentions that the dust activity season occurs from June to October at $20°$- $25°$ S (northern Argentina), and from October to February at $40°$- $45°$ S (southern Argentina). Typically, a diurnal maximum is observed in the afternoon hours.

In the Puna Andean plateau ($22°$- $28°$ S, northwestern Argentina), dry salt basins and lake beds are important dust sources (Middleton, 1986). The dust activity intensifies during the austral winter in this region due to the strong winds resulting from the presence of the subtropical jet stream. In winter, when there is a large temperature gradient between the Tropics and the southern polar region, the jet stream becomes stronger, reaching its northernmost position with its core at $30°$ S (Garreaud, 2009). Winds prevailing in middle and upper troposphere favour a dry winter climate over the Puna because the Andes block

the westerly flow from the Pacific Ocean (Gaiero et al., 2013). As a result, significant dust storms are observed in the region.

## 3 Modeling strategy

Our modeling strategy aims to properly represent the emission process and the subsequent transport of mineral dust and volcanic ash over complex terrain. In the present study, the modeling follows three consecutive steps: (i) Numerical Weather Prediction (NWP) model run, (ii) calculation of dust emission rates, and (iii) dispersal model run.

In the first step, we run the WRF-ARW meteorological model with three nested domains in order to downscale the physical properties of the atmosphere to a spatial resolution of $2 \, \mathrm{km}$ to obtain a reliable background to drive the dust model. The dust emission rates are calculated from the downscaled WRF atmospheric and soil fields. The sensitivity of simulated mineral dust emission to meteorological field resolution is evaluated at this stage. Finally, the evolution of the dust cloud is modeled with the dispersal model.

## 3.1 Meteorological Model

In this study, the meteorological fields were simulated with the ARW core of the Weather Research and Forecasting (WRF) model (*version 3.4*). A detailed description of the WRF-ARW is available in Skamarock et al. (2008). The WRF model configuration used is presented in Table 1.

We used 6-hourly ERA-Interim re-analysis data with spatial resolution of approximately $0.75° \times 0.75°$ and 60 vertical levels

from ground to $0.1 \, \mathrm{hPa}$, obtained from the ECMWF (European Center for Medium-range Weather Forecasts) (Dee et al., 2011)



as initial meteorological field for the WRF domains and boundary condition for the coarsest domain. Friction velocity and soil moisture content are relevant fields for the dust model obtained from the WRF output (see section 3.2.1).

The experiments are conducted with a local closure PBL scheme implemented by Mellor-Yamada-Janjić (MYJ) (Janjic, 1994). The MYJ scheme runs in concurrence with the Eta surface layer scheme based on the similarity theory (Monin and Obukhov, 1954; Janjic, 1996), and provides the friction velocity, $u_*$, for the PBL scheme.

The representation of the land processes is achieved by coupling the WRF model with a land surface scheme, which provides heat and moisture fluxes over land surface to the parent atmospheric model. We use the Noah Land Surface Model (LSM), a 1-D column model with the soil temperature and moisture in four soil layers, with thicknesses of 10, 30, 60, and 100 cm. The Noah LSM predicts the volumetric soil moisture in every layer using the Richard's equation.

## 3.2 Dispersal Model

The simulations were conducted coupling the WRF model to the FALL3D dispersal model, *version 7.0* (Costa et al., 2006; Folch et al., 2009). FALL3D is an Eulerian model for transport and deposition of volcanic ash particles. The model solves the advection-diffusion-sedimentation equations in a curvilinear terrain-following coordinate system using a second-order Finite Differences explicit scheme. The FALL3D model follows an off-line strategy, i.e., a meteorological model provides the meteorological fields a priori.

The diagonal components of the eddy diffusivity tensor, required for the evaluation of kinematic turbulent fluxes, have been parametrized using the similarity theory option (Ulke, 2000) for the vertical component and the CMAQ (Community Multiscale Air Quality) model option (Byun and Schere, 2006) for the horizontal diffusion. The parametrization of the Ganser model (Ganser, 1993) is used to compute the particle settling velocity.

### 3.2.1 Dust emission scheme

The emission rate of windblown dust depends on the transfer of the wind energy to the erodible surface (Marticorena and Bergametti, 1995). Usually, the dust flux is modeled using the friction velocity ($u_*$) concept, a velocity scale for the turbulent fluctuations near the surface (Stull, 1988). It is defined as the square root of the ratio of surface shear stress to atmospheric density. The minimum friction velocity necessary for wind erosion to occur is called threshold friction velocity ($u_{*t}$).

The motion of soil particles by wind depends on particle size and wind speed. Models of dust emission distinguish two major types of grain motion (Greeley and Iversen, 1985). The finest particles ($\lesssim 60\,\mu m$) can remain in the atmosphere for long times and be transported thousands of kilometres, and their trajectories are substantially affected by turbulence. This class of movement is called *suspension*. Particles with diameters of $\sim 70\text{-}500\,\mu m$ follow a smooth trajectory path, mostly unaffected by turbulence (Kok et al., 2012). Such trajectories define a motion called *saltation*.

The Fall3D model in resuspension mode (Folch et al., 2014) includes three different emission schemes (of increasing sophistication) that are applied to the case of volcanic ash, although they have been developed theoretically or calibrated experimentally for mineral dust: (i) Westphal scheme (Westphal et al., 1987), (ii) Marticorena and Bergametti scheme (Marticorena




and Bergametti, 1995; Marticorena et al., 1997), and (iii) Shao scheme (Shao et al., 1993; Shao and Leslie, 1997; Shao and Lu, 2000).

In this work, we use the Shao scheme as is described by Folch et al. (2014) with minor modifications. This parametrization depends crucially on deposit characteristics such as particle size distribution, thickness, and particle density, and requires reliable input data. The theory for dust emission by Shao et al. (1993) is based on a hypothesis about the energetics of dust emission and considers particle-particle interaction by saltation bombardment (see Folch et al., 2014, for more details). The parametrizations used to model the dust fluxes are detailed below.

**Vertical flux**

The experiment confirms that the saltation bombardment is the principal mechanism for the ejection of dust particles and the influence of aerodynamic lift on dust is insignificant under realistic meteorological conditions (Shao, 2008). Therefore, some theoretical models have assumed that the vertical flux of dust emitted is proportional to the horizontal saltation flux (Marticorena and Bergametti, 1995).

The wind tunnel experiments conducted by Shao et al. (1993) indicate that the vertical flux of particles of size $d$ is proportional to $Q(d_s)$, the horizontal flux of saltating particles of size $d_s$:

$$F_V(d, d_s) = \frac{\alpha(d, d_s)}{u_{*t}^2(d)} Q(d_s), \quad d_s \geq d \tag{1}$$

where $\alpha$ is approximately (Shao and Leslie, 1997)

$$\alpha(d, d_s) = [0.6 \log(d_s) + 1.6] \exp(-140d) \tag{2}$$

with $d$ and $d_s$ in milimeters. We used the vertically integrated horizontal (saltation) flux expression suggested by Owen (1964):

$$Q(d) = \frac{c\rho u_*^3}{g} \left[ 1 - \left( \frac{u_{*t}(d)}{u_*} \right)^2 \right], \quad u_* \geq u_{*t} \tag{3}$$

where $c$ is a constant of order unity, $\rho$ is air density, and $g$ is gravitational acceleration.

The dust flux can be estimated from the particle grain size distribution $p(d)$ at each point of the deposit with (Shao and Leslie, 1997):

$$F_V(d) = \sum_{d_s=d}^{d_s=d_{max}} \frac{\alpha(d, d_s)}{u_{*t}^2(d)} p(d) p(d_s) Q(d_s) \tag{4}$$

The source term for the dispersion model was calculated using Eq. (4) for particle sizes d<60 μm in order to take into account only the suspended particles.

**Threshold friction velocity**

The threshold friction velocity represents the resistance of the surface against wind erosion. The most important environmental factors determining $u_{*t}$ are: (i) soil particle size, $d$, (ii) soil moisture, $w$, and (iii) non-erodible surface roughness elements.



In this study, the threshold friction velocity $u_{*t}$ is expressed as:

$$u_{*t} = u_{*t}^o(d) f(w) \tag{5}$$

where $u_{*t}^o(d)$ is the threshold friction velocity on a dry bare soil, and $f(w)$ is a moisture correction. Note that it is not considered the effect of the roughness elements on the threshold friction velocity.

In the case of $u_{*t}^o$, we use the expression for spherical particles loosely spread over a dry and bare surface derived by Shao and Lu (2000):

$$u_{*t}^o(d) = \sqrt{0.0123 \left( \frac{\rho_p g d}{\rho} + \frac{\gamma}{\rho d} \right)} \tag{6}$$

where $\rho_p$ is the particle density and $\gamma = 3 \times 10^{-4} \, \mathrm{kg \, s^{-2}}$.

For the moisture correction, $f(w)$, we used the following empirical expression obtained by Fécan et al. (1999):

$$f(w) = \begin{cases} 1 & w \leq w' \ (\text{dry soil}) \\ \sqrt{1 + 1.21(w - w')^{0.68}} & w > w' \ (\text{wet soil}) \end{cases} \tag{7}$$

where $w$ is the gravimetric soil moisture in percent. The maximum amount of adsorbed water, $w'$, depends on the soil texture. It is negligible for sands and increases with the soil clay content. In this work, $w' = 1\,\%$ is assumed.

## 4   The dust outbreak on 13 June 2015

Episodes of strong dusty winds in Fiambalá throughout 2015 were reported by local media in July, August, September, and October. Dusty winds with gusts up to 90 km/h caused a broad range of damages and complications such as trees down, power outages by falling utility poles, roofs blown off, fires, road transport disruption due to poor visibility and widespread crop injury. Remarkably, local media did not report any event on 13 June.

Prevailing winds in the region are southerly and southeasterly, although in austral winter, strong warm and very dry northerly and northwesterly winds, often carrying dust, are occasionally observed. *Zonda* wind is the regional term for this kind of wind that often occurs on the eastern slope of the Andes, in Argentina. When intense winds in a stable environment reach a mountain range they can produce mountain waves, and descending warm and dry leeward winds. These kinds of winds are usually named depending on the region that they occur, for example: *foehn* in central Europe (Germany, Austria, Switzerland) (Fleagle, 1950; Klemp and Lilly, 1975), *chinook* in Canada and the United States east of the Rocky Mountains (Brinkmann, 1974), *Canterbury-nor'wester* in New Zealand (Lamb, 1974), *berg wind* in South Africa (Lindesay and Tyson, 1990), and *Zonda wind* in Argentina (Norte, 1988) to the east of the Andes Mountain Range (Seluchi et al., 2003). Severe cases of the Zonda near the Andes often occur in winter and spring seasons. Warm, dry and gusty winds experienced during Zonda events can lift and transport large amounts of dust as they reach the east side of the Andes (Allende et al., 2012; Norte, 2015).

Observations on 13 June exhibit the signatures of a Zonda episode in the study region. Figure 2 shows measurements of temperature ($T$), dew point temperature ($T_D$), and surface wind speed from an automatic weather station from the National



Weather Service of Argentina at Catamarca Airport (27°64'36" S-66°35'37" W; altitude: 2,308 m), 200 km from the Fiambalá Basin (see Fig. 1). The automatic weather station measured since 03:00 UTC a strong drying until 20:00 UTC, and a strong increase in wind speed, with a maximum speed of almost 35 kt in the NW direction registered around 15:00 UTC (Fig. 2). According to the GOES-13 water vapor imagery, the dry signature associated with strong subsidence at the lee side of the

Andes mountains in northwestern Argentina is observed during 13 June in accordance to the Zonda event (see the animation in the supplemental material). Particularly, the image at 15:00 UTC of 13 June shows a dry zone and a series of trapped lee waves.

## 4.1 Satellite Imagery

A dust plume was identified using the RGB dust product from the Spinning Enhanced Visible and InfraRed Imager (SEVIRI)

instrument, on board the Meteosat Second Generation (MSG) satellites. The dust product is RGB (Red-Green-Blue) composite based upon infrared channels. It allows detecting suspended dust during both day and night. The RGB composite is produced using the band differences IR12.0-IR10.8 (on red), IR10.8-IR8.7 (on green), and the IR10.8 channel (on blue) (Banks and Brindley, 2013). In the RGB combination, dust appears pink or magenta, a result of the high contrast between the brightness temperatures at the 12 μm and 10.8 μm channels. Thick, high-level clouds have red-brown tones and thin high-level clouds

appear very dark (nearly black).

Figure 3 shows the satellite imagery on 13 June and early morning hours of 14 June. The dust outbreak affected the northern part of Argentina with dust aerosols from the Fiambalá Basin detected from 14:00 UTC by the RGB dust product. The emitted dust was initially transported southeastwards (Fig. 3) over Catamarca province (see Fig. 1). Subsequently, westerly winds advected the dust aerosols towards the East of Argentina over the provinces of Santiago del Estero and Santa Fé. During

the early morning hours of 14 June the more diffused cloud became difficult to identify. For convenience, the area with dust detection is surrounded by dashed line contours. When the cloud is more diluted and the detection becomes more difficult, the cloud tracking is conducted by comparing images at different times to determine the dust contour.

It should be noted that the images are characterized by meteorological clouds dominating northern Argentina (dark spot on the top of the figures).

## 4.2 Reanalysis dataset - Synoptic situation

Figure 4 shows the synoptic situation on 12 June and 13 June, both at 12:00 UTC, according to reanalysis data from ECMWF (Dee et al., 2011). The green shaded in Fig. 4 denotes the region with wind speeds at 500 hPa above 24 m s$^{-1}$. On 12 June, at 500 hPa a cyclonic center located at 85° W over the Pacific Ocean, propagates northeastwards (Fig. 4a). A surface cyclone centered around 55° S, 60° W, over the Atlantic Ocean, and an associated cold front that crosses along the south of Patagonia

was observed. In addition, the frontal cloudiness can be observed in satellite images (see the animation in the supplemental material). The following day (13 June) at 12:00 UTC the surface cyclonic center has propagated eastwards, the frontal zone reached the north of the Patagonia region and a mid-level well-developed trough, located over the Pacific ocean, with strong horizontal pressure gradient produced an increase in horizontal wind speeds over the zone of study (Fig. 4b).





Prevailing westerly winds in the upper troposphere intensify on 13 June above the region of study regarding the previous days because the subtropical jet stream shifts equatorward. In Fig. 5 the Fiambalá location is marked by a red circle in a meridional cross-section of zonal wind on 13 June at 18:00 UTC. The subtropical jet stream core at 250 hPa is located close to latitude 30° S, above the region of study.

In summary, the synoptic key features that produce Zonda events (Seluchi et al., 2003; Norte, 2015) are present in this case of study: a mid-level trough west of the Andes, a surface cyclone southeast of the barrier near the extreme of the continent, a jet stream located over the region of study, and the frontal zone over the northern regions of Patagonia.

### 4.3  Reanalysis dataset - June time series

The ERA-Interim dataset for June 2015 was analysed to ascertain the exceptional conditions that occurred on 13 June. Figure 6
shows the time series throughout June for relevant variables at Fiambalá location. The wind speed at 500 hPa (Fig. 6a) shows a peak on 13 June (see the reference line at 18:00 UTC). Figure 6b suggests that dry soil conditions could create favourable conditions for dust mobilisation, as the lowest volumetric soil moisture in June occurred on 13th.

Remarkably, the surface wind speed on 13 June did not exceed the June 2015 average (Fig. 6c), suggesting that high-resolution simulations are required to explain the dust outbreak. In the next section is carried out a thorough study of the key
role played by terrain-induced mesoscale disturbances of the large-scale prevailing flows.

## 5  Results and discussion

### 5.1  Domain configuration

The WRF model was integrated up to 54 h with the initial conditions at 00:00 UTC on 12 June, considering the first 12-h period for model spin-up. In the vertical, 60 levels were set with model top at 50 hPa.
Figure 7a shows the domain configuration. It consists of three two-way nested domains with grid sizes of 18, 6 and 2 km for domains d01, d02 and d03, respectively. The coarsest domain covered the northern Argentina. The 6 km domain covered northwestern Argentina including the complex topography of the Andes Mountain Range. The finest 2-km domain covered the Fiambalá Basin. In each case, the domains are centered at 27.5° S, 67.7° W, i.e. the coordinates of Fiambalá, approximately.

The topography data was obtained from the U.S. Geological Survey (USGS) global 30 arc-s elevation dataset. The terrain
heights around the Fiambalá Basin for the three nested domains are shown in Fig. 7b, c, and d. The solid lines in these figures delimit the Fiambalá Basin and define the area of potential dust sources in our modeling strategy.

In the figures, the importance of the horizontal resolution for a proper representation of the orographic features of the region becomes evident. Particularly, Fig. 7d shows an extremely abrupt terrain for the 2-km domain with variations in terrain height between 6000 m and 700 m.
Due to strong vertical wind velocities in steep terrain, the model develops numerical instability for a large enough time-step. For WRF-ARW, time-step (in seconds) should typically be about $6 \times \Delta x$, with the grid spacing, $\Delta x$, in kilometers. However,




we found that time-steps greater than $\sim 3 \times \Delta x$ triggered violations of the Courant-Friedrichs-Lewy (CFL) condition. In this work, we used the adaptive time-step option to avoid CFL instabilities.

The WRF-ARW fields (on a Lambert Conformal projection) were interpolated onto the FALL3D regular lat-lon grid by bilinear interpolation. The FALL3D model was run using two domains: (a) the fine domain with $0.01°$ grid resolution, centered over the Fiambalá Basin, and (b) the coarse domain with $0.1°$ grid resolution, covering the northern part of Argentina. The fine resolution domain spans from $28°$ S to $27°$ S in latitude and from $68°$ W to $67.3°$ W in longitude, whereas the coarse domain spans from $32°$ S to $22°$ S in latitude and from $72°$ W to $56.8°$ W in longitude. In both cases 57 vertical levels up to $8.5\,\mathrm{km}$ a.g.l. (above ground level) were used. We run FALL3D on the fine and coarse domains to simulate the short- and long-range transport of mineral dust, respectively.

## 5.2 Zonda episode

According to simulations, the surface wind intensified significantly on 13 June. Figure 8 shows horizontal wind vectors at $10\,\mathrm{m}$ above the ground simulated by the WRF model for the 2-km innermost domain. Light winds dominated the Fiambalá Basin during 12 June (Fig. 8a). On 13 June, the presence of a Zonda wind caused strong downslope winds in northern Fiambalá Basin (Fig. 8b). In the figure we identify the geographic location $P$ ($27.25°$ S, $67.7°$ W) in northern Fiambalá Basin, as a reference. The maximum mean wind speed at $P$, reached at 15:00 UTC on 13 June, was about $20\,\mathrm{m\,s^{-1}}$ blowing from the NE direction.

The mountain waves that accompany Zonda events can be recognized in this case. Figure 9a shows a vertical cross section of equivalent potential temperature and vertical wind velocity through the Andes at the latitude of $P$ ($27.25°$ S). The theta-e contours show an unstable layer near surface on the windward side. At the top of the mountain a stable layer was observed that will remain during the whole period of study. Above the Fiambalá Basin ($P$ in Fig. 9a), on the lee side, the troposphere is less stable with strongly tilting isentropes and descending airflow at all levels. The Skew-T diagram at $P$ shows other typical characteristics of a Zonda episode (Fig. 9b): a very dry air column at all levels and a nearly dry adiabatic lapse rate at low levels, suggesting subsidence reaches the surface (Seluchi et al., 2003). The emergence of a descending air mass becomes evident in the time series of the vertical wind speed ($W$) at $P$ (Fig. 9c). According to the simulations, there is a vertical downward wind speed from 13 June at 00:00 UTC until 14 June at 00:00 UTC, approximately. The time series of the temperature ($T$) and the dew point temperature ($T_D$) at $P$ show a strong drying, compatible with the presence of a Zonda wind, throughout 13 June. Notice that the WRF time series for $T$ and $T_D$ are similar to measurements at Catamarca station (Fig. 2).

### 5.3 Source strength

The emission rates are obtained as a pre-processing step by FALL3D dispersal model using the emission scheme proposed by Shao and Leslie (1997) (thereinafter "Shao scheme"). According to the Shao scheme, Eq. (4), the emission rate depends on the particle grain size distribution $p(d)$ of the deposit. We used a log-normal distribution discretized along 14 classes ranging from $\phi = 0$ to $\phi = 8$ ($\phi = -\log_2(d)$, with $d$ in mm). The parameters of the distribution, obtained from field measurements, are: mean, $\overline{\phi} = 3\phi$, and standard deviation, $\sigma_\phi = 1.5\phi$ (J. G. Viramonte, personal communication, 2015). The density of particles varies linearly with $\phi$ between 1.2-2.2 $\mathrm{g\,cm^{-3}}$.



The elevation at which dust aerosols are injected into the atmosphere has a strong influence on the subsequent transport. The time-dependent source term is obtained assuming an uniform vertical distribution of the emitted mass up to a fixed injection height. In this study, the mass is injected between the two lowest levels (10 m a.g.l. and 160 m a.g.l.).

The WRF output fields required for the emission scheme are friction velocity and soil moisture for the upper layer (at 10-
cm depth). Figure 8 shows that the friction velocity for the innermost WRF domain (2 km) is strongly correlated with the orographic features of the region, with the largest friction velocities occurring in steep terrain.

Figure 10 shows the grid emitted mass throughout the 54-h simulation, computed from the 2-km WRF fields. The dust emission occurs predominantly in the north of the Fiambalá Basin due to the strong downslope winds of the region. Figure 11 shows the time series of the dust emission rate (emission flux spatially integrated) for the 2-km, 6-km, and 18-km WRF fields.
According to the emission rate driven by the 2-km $u_*$, the dust emission occured on 13 June during the presence of the Zonda wind episode. The model predicts a maximum emission at around 15:00 UTC. The dust emission fluxes are very sensitive to horizontal resolution: the source strength computed with the 6-km and 18-km meteorological fields decreases significantly. In these cases, a reduced duration of the dust outbreak is predicted by the model, with the maxima remaining at 15:00 UTC. The total mass emitted for the entire study period for each domain were $1.7 \times 10^9$ kg (2 km), $0.48 \times 10^9$ kg (6 km), and $0.17 \times 10^9$ kg
(18 km). These results highlight the importance of a proper representation of the orographic features.

The gray-shaded area in Fig. 11 represents the time window with satellite detection of dust around the source area. The dust cloud is detected by the SEVIRI RGB dust product from 14:00 UTC. According to simulations, the emission occurred before satellite detection.

It should be noted that errors can exist due to the uncertainty in the area of potential sources of dust. Since the simulations
depend strongly on the availability of local dust sources, a complete data set of regional soil properties is required for improvement of the results. Additionally, there are errors associated with the WRF-ARW output fields, because the scales involved in the mesoscale model are not representative of the spatial scales pertinent to the dust emission processes (i.e., aeolian scales). In fact, the WRF surface layer scheme is based on the Monin-Obukhov similarity theory (Monin and Obukhov, 1954). According to this theory, the calculation of $u_*$ depends on the aerodynamic roughness length, $z_0$. As noted Darmenova et al. (2009),
mesoscale model roughness lengths operate at scales unsuitable for dust emission parametrizations. In this case, an alternative approach to compute the friction velocity would be advisable. For example, Darmenova et al. (2009) suggested recalculate $u_*$ using an external data set of $z_0$ instead of the aerodynamic roughness length from the mesoscala model. Furthermore, the 10-cm depth soil moisture appears to be unrepresentative of the soil surface conditions involved in the dust emission processes.

### 5.4 Short-range transport

In this section, the FALL3D results corresponding to the short-range simulations (0.01° domain) are analysed in order to study the spatial distribution of dust concentration over the Fiambalá Basin. Figure 12 shows a meridional cross section along P longitude (67.7° W) at different times.

At 11:00 UTC on 13 June, mineral dust is emitted from the northern part of the Fiambalá Basin, between 27.2° S and 27.1° S (Fig. 12a). The dust emission region can be recognized by the particle column mass peak (red solid line) and the large values





of dust concentration. Aerosol particles remain confined to near-ground layers north of latitude 27.5° S and tend to be removed from the atmosphere by settling. An abrupt reduction of the near-ground concentration of dust occurs south of latitude 27.5° S, where dust aerosols are injected into greater heights.

At 16:00 UTC on 13 June a second peak in the particle column mass profile of Fig. 12b denotes the emergence of new
emission sources between 27.4° S and 27.3° S. In this case, it is observed a similar situation as that in Fig. 12a: in the northern part of the Fiambalá Basin dust particles are distributed over the lowest atmospheric layer, whereas in the central and southern regions the vertical distribution of dust aerosols extends up to 5-6 km.

The spatial distribution pattern of the dust concentration can be explained from Fig. 13, where a meridional cross section at 16:00 UTC of vertical wind velocity is shown. A descending air mass dominates the northern region of the Fiambalá Basin
leading to an accumulation of particles in the lower layers and creating favorable conditions for dust emission due to the strong downslope winds in surface. On the other hand, Fig. 13 shows the presence of ascending airflows driven by horizontal convergence between 27.5° S and 27.4° S (see also Fig. 8) with vertical velocities greater than $6\,\mathrm{m\,s^{-1}}$. Dust aerosols are injected to greater heights in this region according to the dust vertical distribution shown in Fig. 12b.

This result suggests that ascending airflow driven by horizontal convergence in the Fiambalá Basin might have been an
important dust injection mechanism. In order to quantify the dust injection height, we define the cloud top height at a specific location as the height at which a concentration of $1\,\mathrm{mg\,m^{-3}}$ is attained (i.e. the highest contour in Fig. 12). Figure 14 shows the time series for the cloud top height averaged over the FALL3D domain. Two mechanisms promote the dust injection into the atmosphere from 12:00 UTC: (a) the ascending air mass intensifies, and (b) the emission occurs close to the surface convergence. The latter mechanism is clearly visible from Fig. 12b. The southernmost peak of the particle column mass profile
denotes a set of dust sources supplying with particles to the ascending air airflow located at ∼ 27.4° S. As a consequence, an important amount of mass can be uplifted and the background column mass over the Fiambalá Basin increases.

According to the numerical simulations, the maximum height is reached at 16:00 UTC and a steep increase of the cloud top occurs between 14:00 UTC and 15:00 UTC, when dust satellite detection begins (gray-shaded area). The elevated height of the dust cloud and the increased column mass background appear to be the conditions that allowed the satellite detection of dust
over the central and southern regions of the Fiambalá Basin. Satellite detection of mineral dust becomes evident between ∼ 27.5° S and ∼ 27.9° S in the SEVIRI image at 16:00 UTC on 13 June, in agreement with the region where the modeled dust cloud reaches its maximum height in Fig. 12b. On the other hand, the low cloud heights prevent satellite detection in the north of the Fiambalá Basin, where the emission occurs and the model predicts the highest concentration values.

## 5.5   Long-range transport

Numerical simulations of long-range transport were performed to model dust cloud spreading over northern Argentina. The results of FALL3D (0.1° domain) are compared with satellite imagery in this section.

Both the meteorological and the dispersal models in our modeling strategy use terrain-following coordinates. As a consequence, simulations over the extremely complex orography of the region represent a challenge for the modeling system. For example, it is recognized that errors in computing the horizontal pressure-gradient force in terrain-following coordinates





may arise in regions of steep terrain (Janjic, 1977; Klemp et al., 2003). The WRF-ARW equations are formulated using a terrain-following hydrostatic-pressure vertical coordinate (Skamarock et al., 2008). Instead, FALL3D model uses a simple terrain-following coordinate system where the horizontal coordinates remain unchanged: $x = X$, $y = Y$, $z \rightarrow Z$, where $Z = z - h(x,y)$, with $h(x,y)$ denoting the topographic elevation (Costa et al., 2006).

We conducted the simulations using two numerical methods to compute the advection terms of the advection-diffusion-sedimentation equation from the FALL3D model (Costa et al., 2006):

$$U\frac{\partial C}{\partial x} + V\frac{\partial C}{\partial y} + W\frac{\partial C}{\partial z}. \tag{8}$$

In the simpler method (default configuration), the terms of Eq. (8) are computed using the wind components provided by the WRF model, and the gradients are computed using points on the same model level. In the alternative method, the vertical wind

velocity is replaced by

$$W \rightarrow W - \left(U\frac{\partial h}{\partial x} + V\frac{\partial h}{\partial y}\right). \tag{9}$$

to more accurately compute the horizontal advection in sloped coordinates. Using this approach the Eq. (8) remains invariant under the transformation of the FALL3D terrain-following coordinate system.

  The modeled particle column mass on 13 June at 23:00 UTC using the default method is shown in Fig. 15. The dust aerosols

remain close to the source region throughout the simulation. In this case, the concentration diminishes steeply through the *Sierra de Ambato* orographic barrier (see Fig. 1). Figure 16 shows the modeled concentration on 13 June at 23:00 UTC in a West-East cross section along the red line shown in Fig. 15. Numerical errors in the advection calculation can lead to a poor performance in sloping terrain using the FALL3D default configuration. Figure 16a shows that the particles tend to remain in the same model level and the dust aerosols are partly "blocked" by the orographic barrier. With the method based on the Eq. (9),

the dust cloud is transported eastward over longer distances downwind the orographic barrier (Fig. 16b), more in agreement with the satellite imagery (see Fig. 3). The results show that the correction based on Eq. (9) improves the performance of FALL3D in sloped terrain. Therefore, it is advisable to use this configuration by default.

  The sequence of particle column mass modeled using the vertical velocity correction, Eq. (9), is shown in Fig. 17. Initially, dust raised into the atmosphere is carried southeastwards by the orographic winds. Subsequently, the uplifted dust reaches

greater heights and the dust cloud is advected eastwards by the mid-level westerly winds. Comparisons among model output and satellite imagery reveal that the modeled dust cloud is further north (see Fig. 3). As a conclusion, these results highlight the difficulty of modeling the transport of aerosols in the extremely complex terrain of the region.

## 6 Conclusions

On 13 June 2015, the Buenos Aires VAAC was informed about a suspected volcanic eruption in the proximity of the Ojos

del Salado-San Buenaventura volcanic lineament. A volcanic ash cloud was detected by the infrared channels of the Spinning Enhanced Visible and Infrared Imager (SEVIRI) instrument, carried on board Meteosat Second Generation (MSG) satellites.





This event triggered a thorough interdisciplinary investigation. As a result, Collini et al. (2015) concluded that the phenomenon was caused by remobilization of ancient pyroclastic deposits from the Bolsón de Fiambalá (Fiambalá Basin).

In this work, we performed numerical simulations of windblown dust using the WRF-ARW/FALL3D modeling system. A successful description of the emission processes and the spatio-temporal distribution of dust concentration represents a challenge for the modeling system due to the the extremely complex orography of the region, with variations in terrain height of about 5000 meters over horizontal distances of the order of $100\,\mathrm{km}$.

The modeling strategy followed three consecutive steps: (i) meteorological model run, (ii) calculation of dust emission rates, and (iii) dispersal model run. In the step (i), the WRF-ARW meteorological model was run using three two-way nested domains. The meteorological fields were downscaled to a spatial resolution of $2\,\mathrm{km}$ to resolve the complex orography of the study region. The results of the numerical simulations are compatible with the presence of a Zonda wind throughout 13 June: formation of mountains waves, a very dry air column on the lee side of the mountain range, and strong surface winds.

(ii) Emission flux: The WRF output fields are required by the FALL3D model to compute the emission rates and the subsequent transport of mineral dust. The dust emission rate was computed using the parametrizations of the Shao scheme implemented in FALL3D with minor modifications. According to the results, during the dust outbreak strong downslope winds affected the northern part of the Fiambalá Basin, creating favorable conditions for the uplift of mineral dust. We showed that the simulated emission rates are very sensitive to horizontal resolution of the meteorological fields. Specifically, the total dust emitted for the entire study period were: 1.7, 0.48, and $0.17 \times 10^9\,\mathrm{kg}$ for the spatial resolutions of 2, 6, and $18\,\mathrm{km}$, respectively.

(iii) Dispersal model run: In order to study the spatial distribution of the modeled dust concentration over the Fiambalá Basin, we performed numerical simulations of short-range transport using the FALL3D model. Most of the particles are concentrated in the northern regions, where the emission sources are located. As a consequence of the intensive descending airflows, with downward velocities of over $7\,\mathrm{m\,s^{-1}}$, the particles are distributed in a thin near-ground layer in these regions. Conversely, dust aerosols were injected up to 5-6 km in the central and southern regions of the Fiambalá Basin according to simulations.

Long-range transport numerical simulations were also performed to model dust cloud spreading over northern Argentina and the simulated vertical particle column mass was compared with the MSG-SEVIRI retrieval product. We tested two numerical schemes: Difficulties arose to simulate transport through orographic barriers with the default configuration of the FALL3D model. An alternative configuration, using a numerical scheme to more accurately compute the horizontal advection in abrupt terrains, substantially improved the model performance.

*Acknowledgements.* L. Mingari thanks CONICET for their PhD fellowship. The WRF-ARW/FALL3D modeling system ran in a server installed at the SMN with funds from the Argentinean project PIDDEF 41/10: "Pronóstico del tiempo para estudios de vulnerabilidad e impacto socioeconómico". The images from SEVIRI were provided by Diego Souza from National Institute for Space Research (INPE), Center for Weather Forecast and Climate Studies (CPTEC), Satellite Division and Environmental Systems (DSA), Brazil. We thank the Servicio Meteorologico Nacional (Argentina) for providing the meteorological data from the weather station at Catamarca Airport. The authors would like to thank Miriam Andrioli of the National Meteorological Service of Argentina for providing information about the study episode. We thank Julia Paegle for her valuable comments on the manuscript.



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



**Table 1.** WRF-ARW configuration

| Parameter | Value |
|---|---|
| *Input data:* | |
| Terrain and land use | USGS |
| Initial and boundary | ERA-Interim |
| Temporal interval of boundary | 6 h |
| *Time control:* | |
| Start | 13 June, 00:00 UTC |
| End | 14 June, 06:00 UTC |
| *Domains:* | |
| Map projection | Lambert Conformal |
| Nesting | Two-way (multiple input files) |
| Spatial resolution | 18, 6 and 2 km |
| Vertical levels | 60 |
| Temporal resolution | Adaptative time step |
| Minimum time step (sec) | $\Delta x$ (km) |
| Maximum time step (sec) | $6 \times \Delta x$ (km) |
| *Physics:* | |
| PBL Scheme | Mellor-Yamada-Janjic |
| Surface | Noah Land Surface Model |
| Surface layer physics | Eta Similarity |
| Microphysics | Eta (Ferrier) |
| Short wave radiation | Dudhia scheme |
| Long wave radiation | RRTM scheme |
| *Dynamics:* | |
| Dynamics | Non hydrostatic |
| Time integration | Runge-Kutta 3rd order |
| Turbulence and Mixing | $2^{nd}$ order diffusion on model levels |
| Eddy Coefficient | 2D Smagorinsky |
| Horizontal Momentum Advection | $5^{th}$ order |
| Vertical Momentum Advection | $3^{rd}$ order |

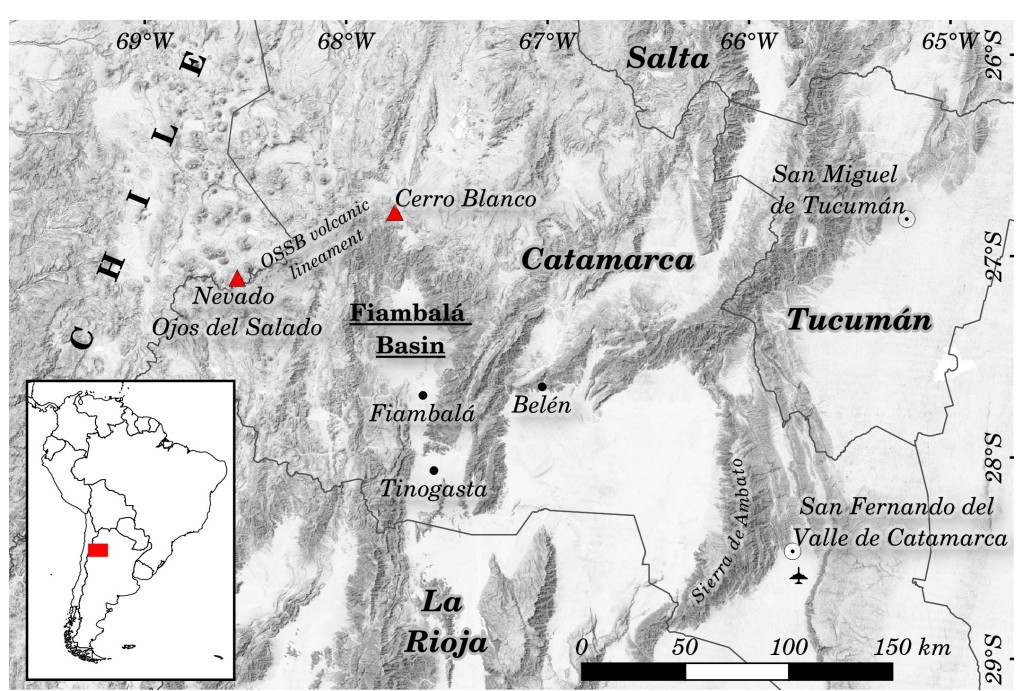

**Figure 1.** Map of the region under study. The triangles indicate the location of volcanoes mentioned in the text. The location of an automatic weather station at Catamarca Airport is shown south of San Fernando del Valle de Catamarca. SRTM, ASTER GDEM is a product of METI and NASA, Imagery GIScience Research Group @ Heidelberg University.



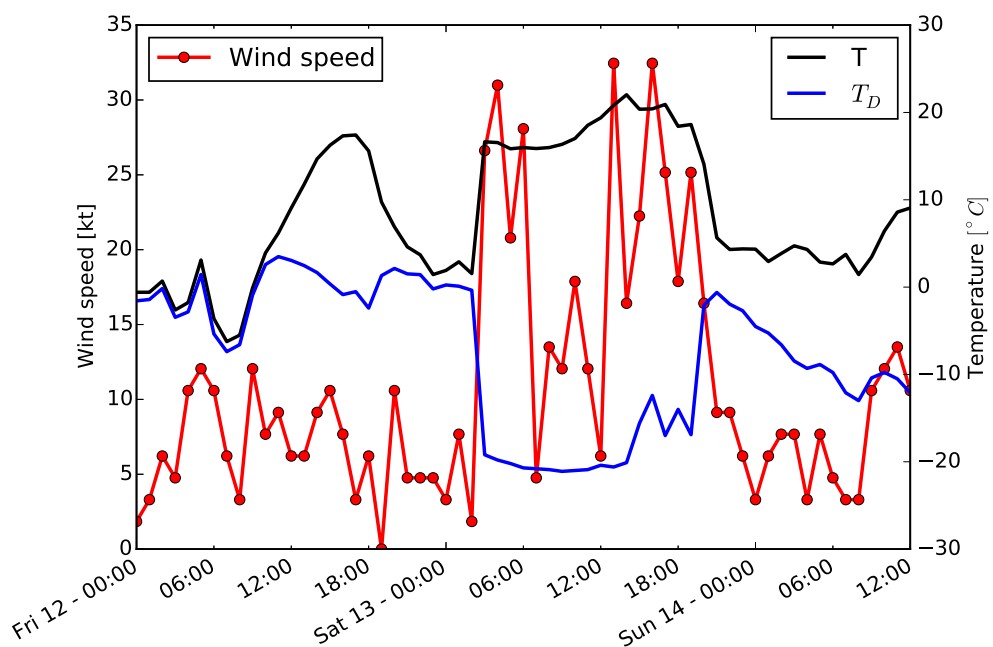

**Figure 2.** Measurements from an automatic weather station from the National Weather Service of Argentina at Catamarca Airport, 200 km from the Fiambalá Basin. Wind speed (red line with circle markers), temperature ($T$, black line), and dew point temperature ($T_D$, blue line). The signatures of a Zonda episode are observed on 13 June.





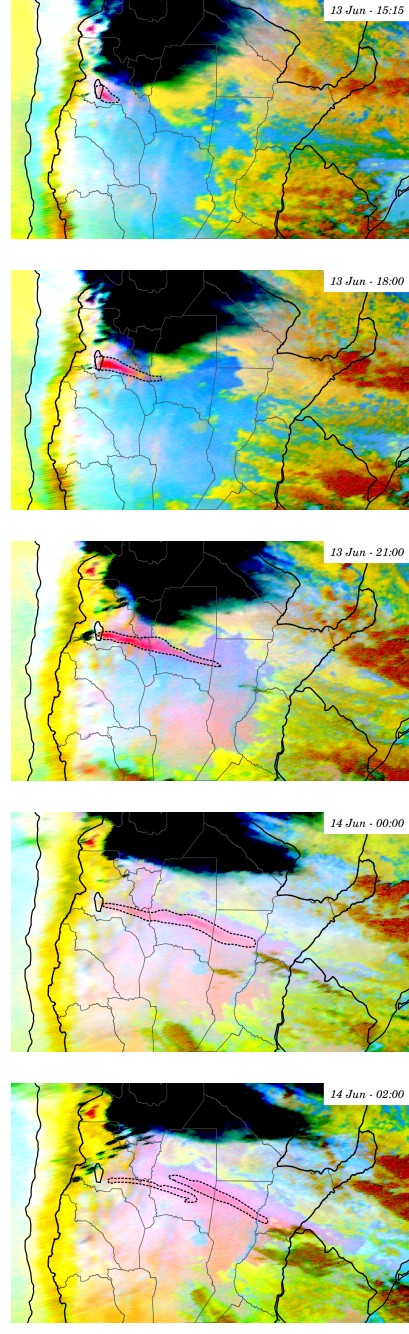

**Figure 3.** Dust detection with the MSG-SEVIRI RGB Product at 15:15, 18:00, 21:00 (13 June), 00:00, and 02:00 UTC (14 June). The dust cloud (dashed line) moved southeastwards across northern Argentina. The dark spots on the top of the figures represent meteorological clouds affecting parts of Argentina, Bolivia, Paraguay and Chile.





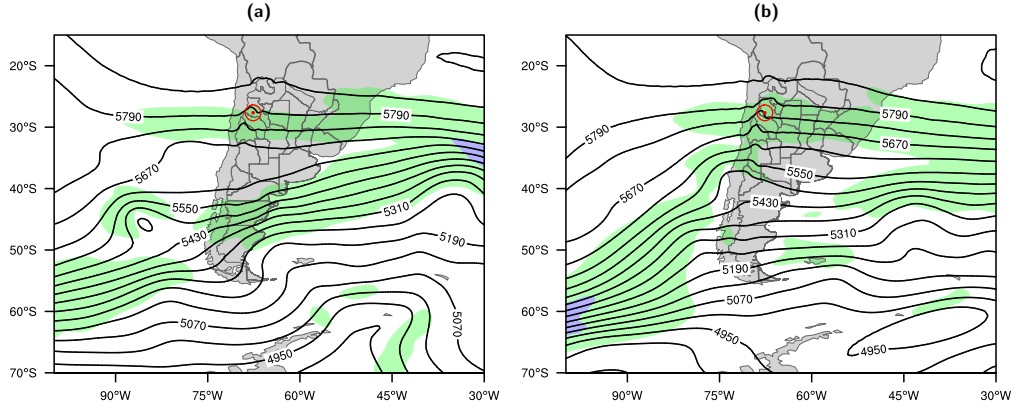

**Figure 4.** Meteorological situation corresponding to 12:00 UTC on 12 June 2015 (a) and 12:00 UTC on 13 June 2015 (b). Solid line: the 500 hPa geopotential height (gpm). Shaded contour: Wind speed $> 24\,\mathrm{m\,s^{-1}}$. Fiambalá location is stated by a red circle. Dataset from ERA-Interim re-analysis.

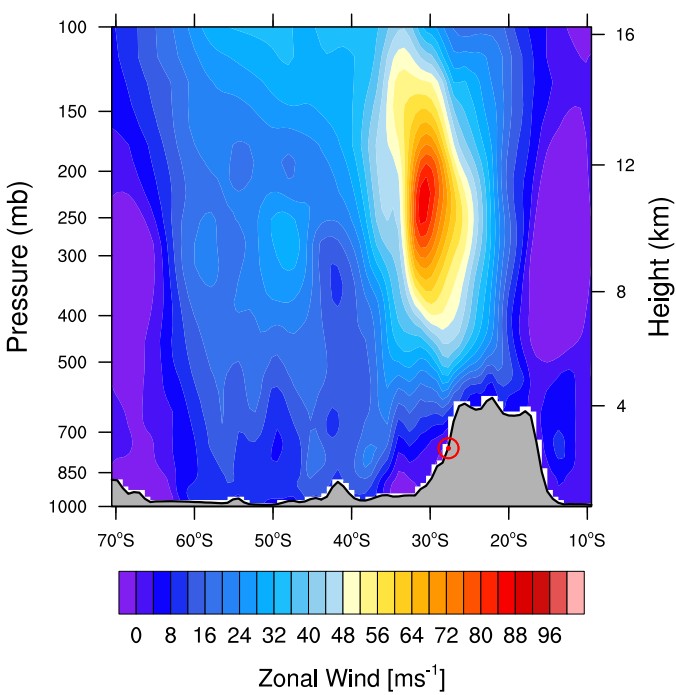

**Figure 5.** Meridional cross-section of zonal wind along the Fiambalá longitude on 13 June at 1800 UTC. The subtropical jet stream core at 250 hPa shifts equatorward, reaching the Fiambalá latitude (stated by a red circle). Dataset from ERA-Interim re-analysis.



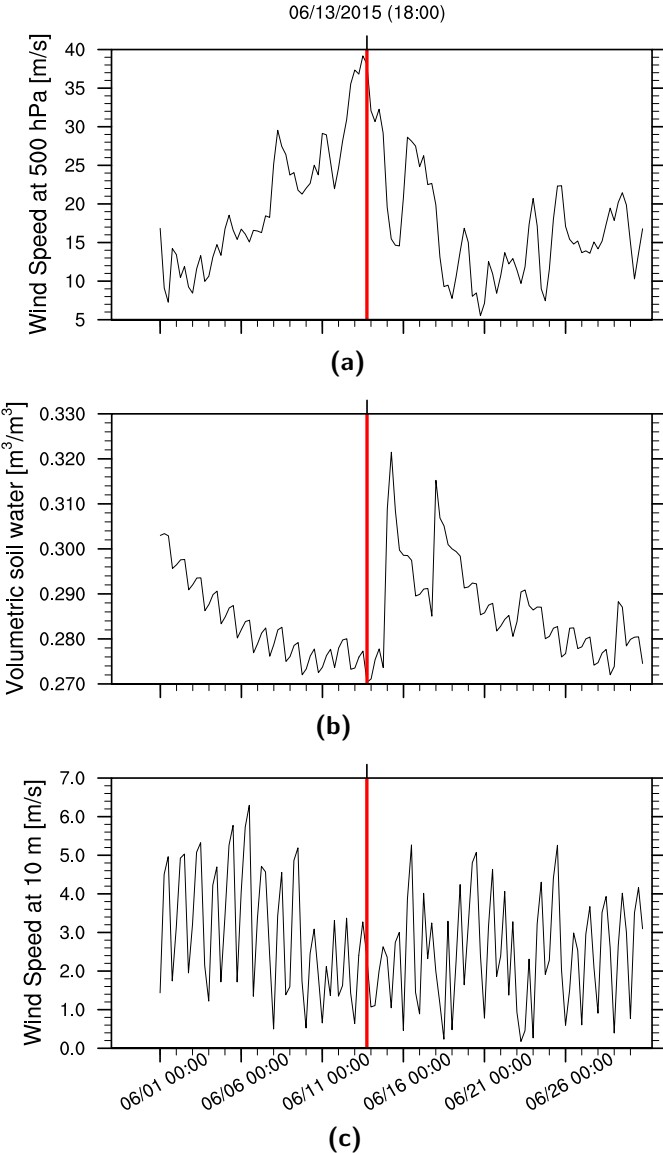

**Figure 6.** Time series at Fiambalá coordinates during June for wind speed at 500 hPa (a), volumetric soil water (b), and wind speed at 10 m (c). The conditions on 13 June at 18:00 UTC (red line) allowed dust to be raised in suspension in the atmosphere. Dataset from ERA-Interim re-analysis.



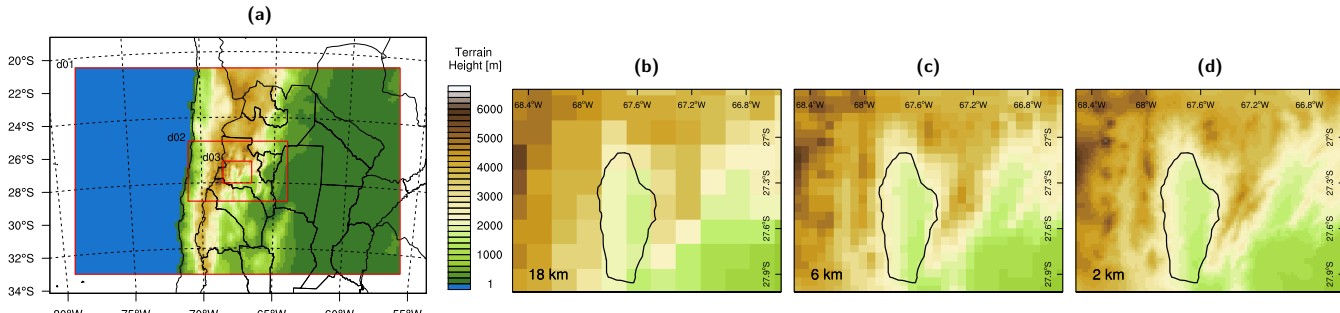

**Figure 7.** WRF model domain configuration. The colours indicate the model terrain height. Three two-way nested domains enclosing the Fiambalá Basin with horizontal resolutions are 18 km, 6 km, and 2 km, from the parent to the innermost domain, respectively (a). Terrain height around the Fiambalá Basin for the 18-km (b), 6-km (c), and 2-km (d) domains. Topography data from the U.S. Geological Survey (USGS).





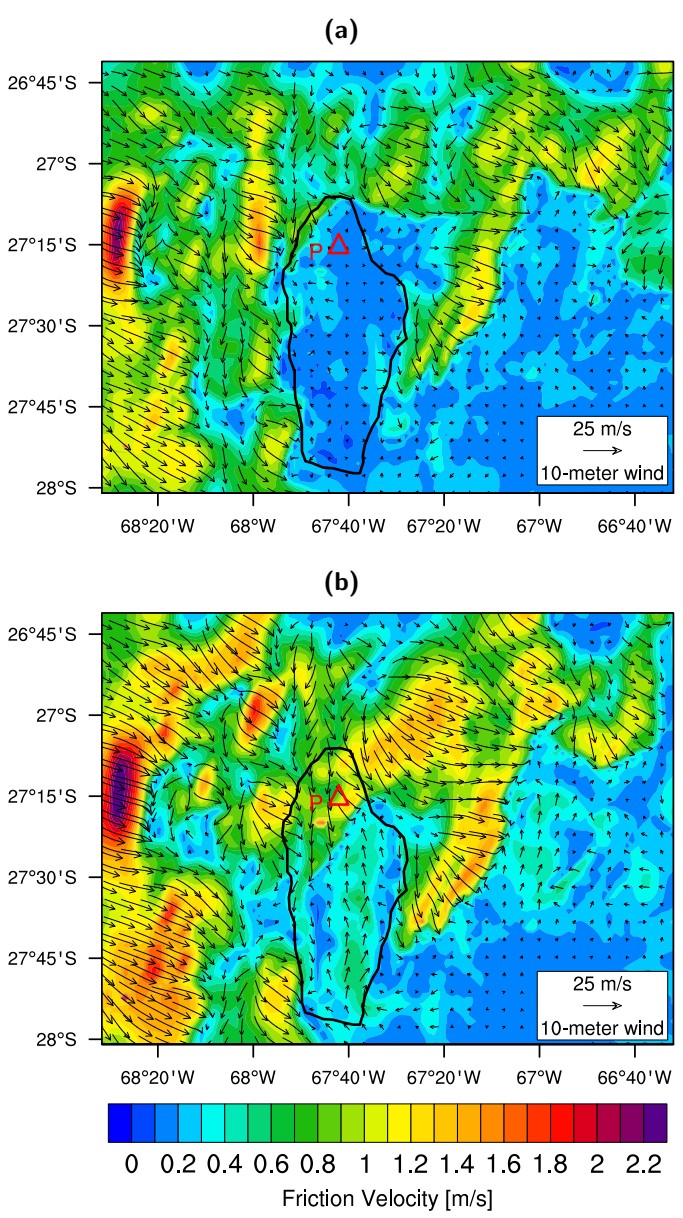

**Figure 8.** Friction velocity contours and wind vectors at 10 metres above the surface simulated by WRF model for the 2-km innermost domain at 15:00 UTC on 12 June (a) and 15:00 UTC on 13 June (b). We identify the geographic location $P$ for future reference.



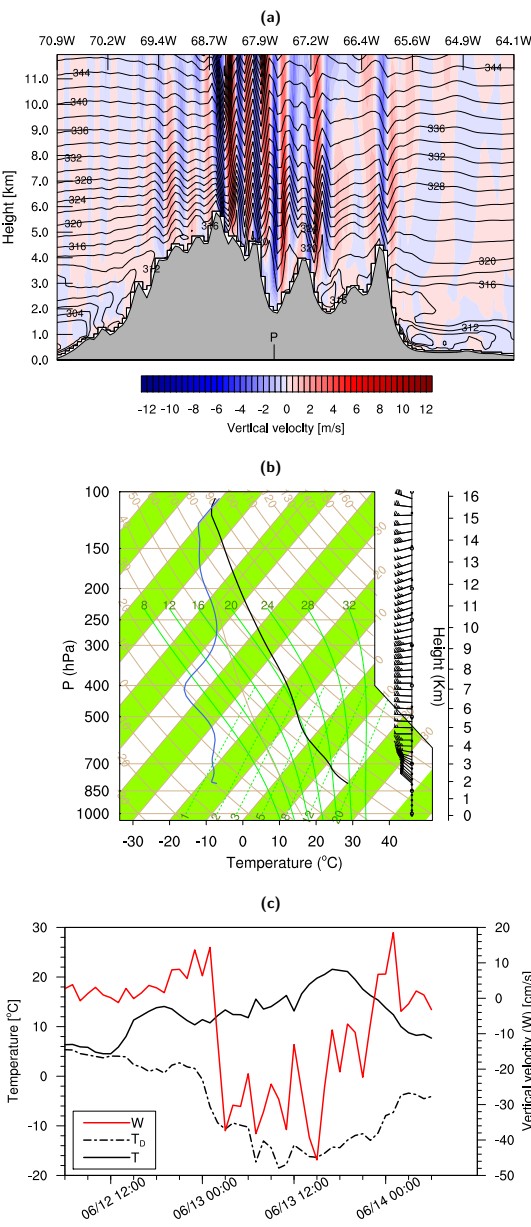

**Figure 9.** Features of a Zonda event in the northern Fiambalá Basin according to WRF-ARW output. (a) Vertical cross section of equivalent potential temperature (K) and vertical wind velocity ($\mathrm{m\,s^{-1}}$) through the Andes along the latitude of $P$ (27.25° S) at 16:00 UTC on 13 June. (b) Skew-T Log-P diagrams at $P$ (27.25° S, 67.7° W) at 16:00 UTC on 13 June. (c) Time series at $P$ for temperature ($T$), dew point temperature ($T_D$), and vertical velocity ($W$). Temperatures at 2 m and vertical wind velocity at the first model level ($\sim 27.5\,\mathrm{m}$).





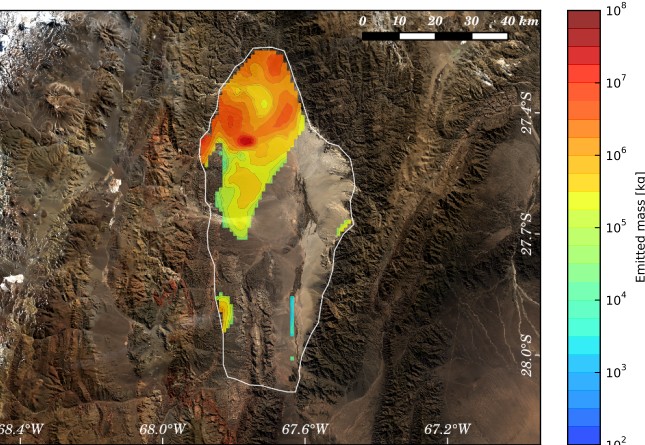

**Figure 10.** Grid emitted mass throughout the 54-h simulation according to the Shao emission scheme using data from the innermost domain d03 (2 km). In the background: Landsat-8 image acquired on 14 June 2015.

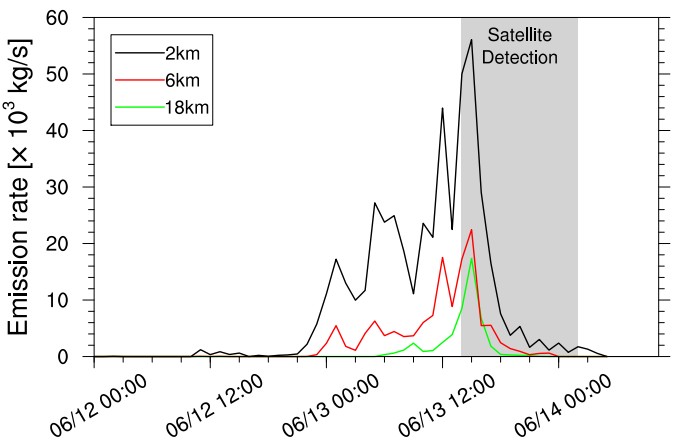

**Figure 11.** Time series for the emission rates (spatially integrated) according to Shao emission scheme. Results are very sensitive to horizontal resolution of WRF meteorological fields.





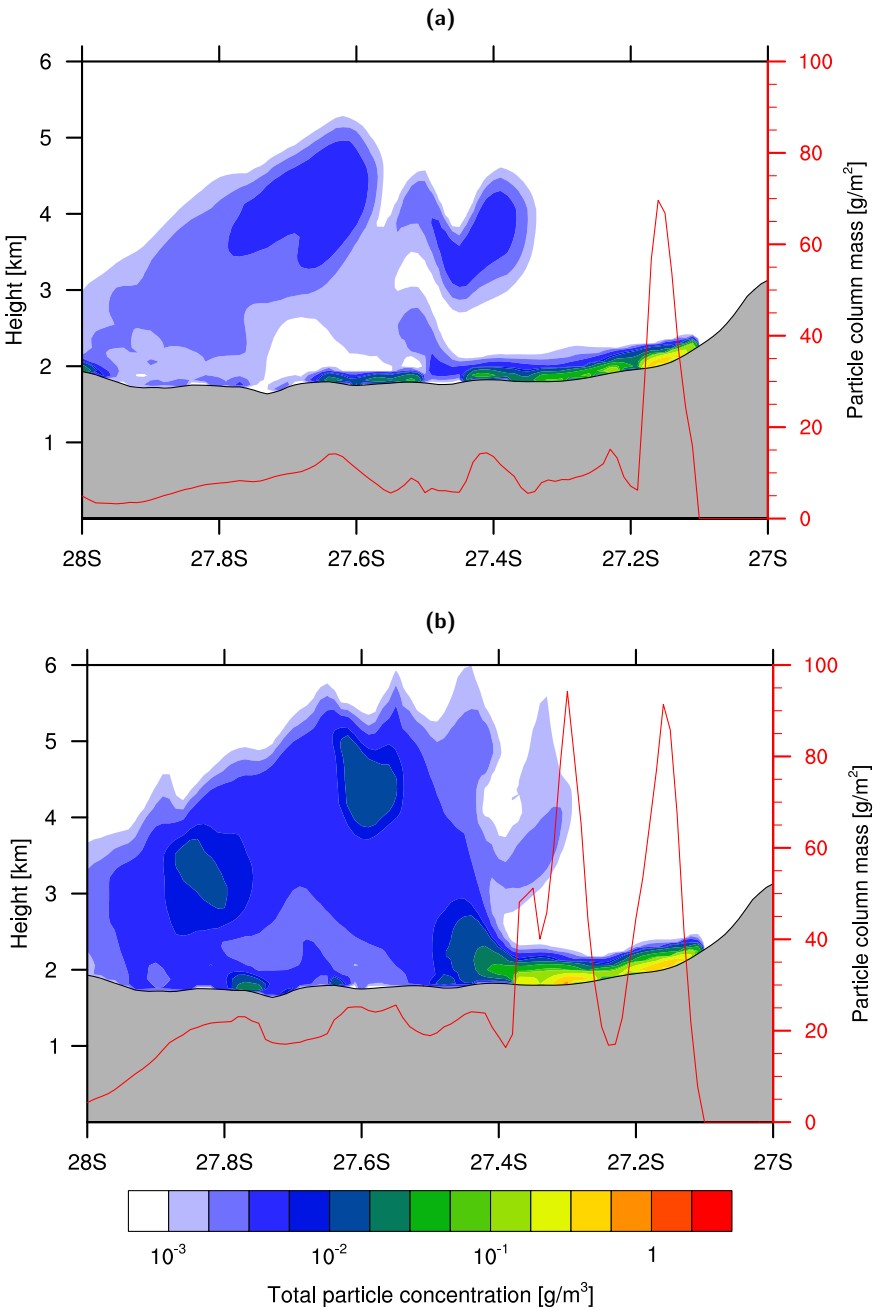

**Figure 12.** Spatial distribution of the modeled dust concentration over the Fiambalá Basin. Meridional cross section along longitude 67.7° W. The red solid line represents the particle column mass profile. The emission sources are centered around the peaks of column mass. (a) 13 June at 11:00 UTC. (b) 13 June at 16:00 UTC.




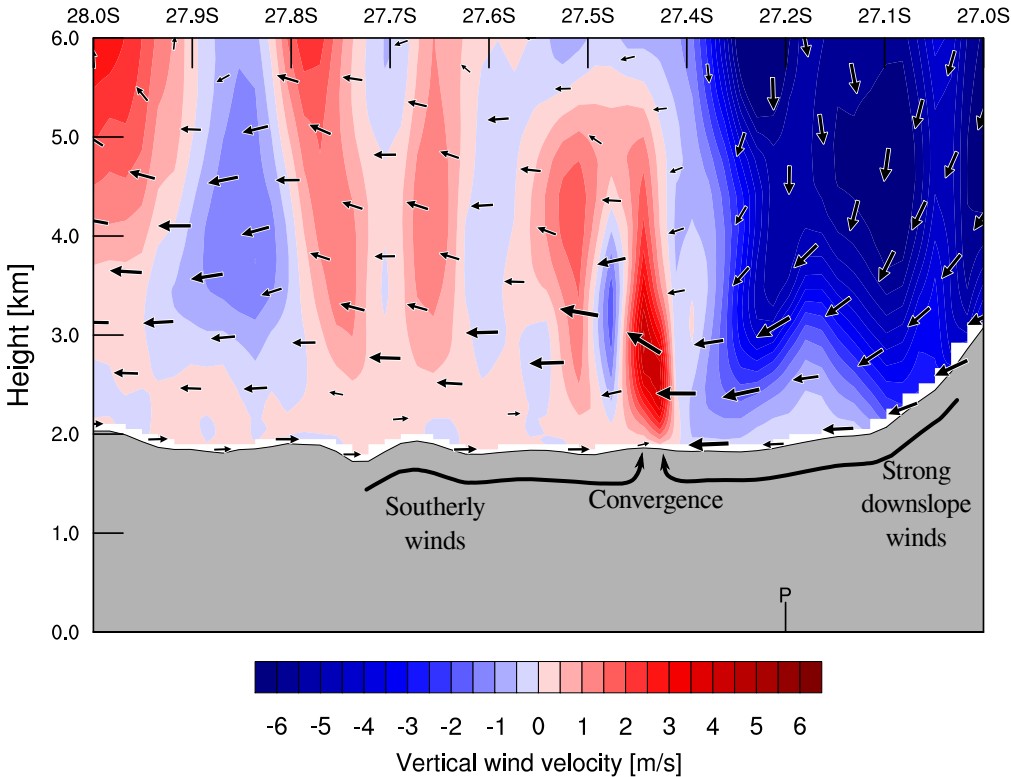

**Figure 13.** Meridional cross section of vertical velocity on 13 June at 16:00 UTC. Longitude: $\sim 67.7^\circ$ W.

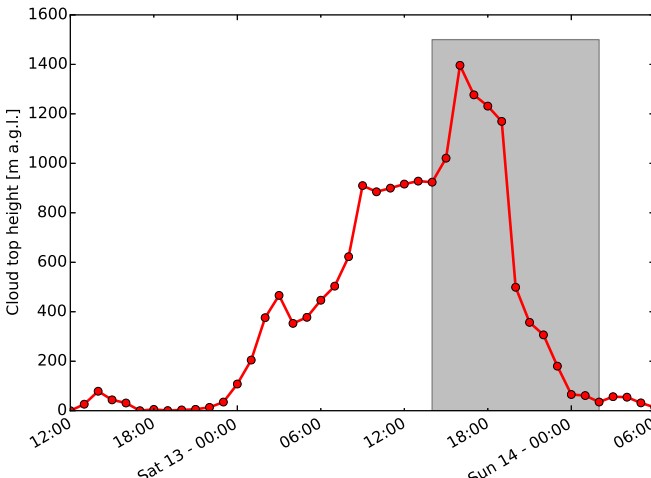

**Figure 14.** Time series for the cloud top height averaged over the FALL3D domain. The gray-shaded area denotes the satellite detection period.





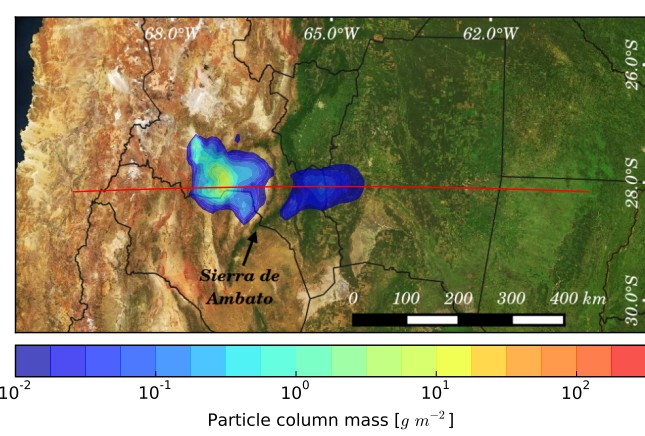

**Figure 15.** FALL3D output column mass on 13 June at 23:00 UTC using the default configuration. The transport of dust aerosols is partly blocked by the orographic barrier *Sierra de Ambato*.



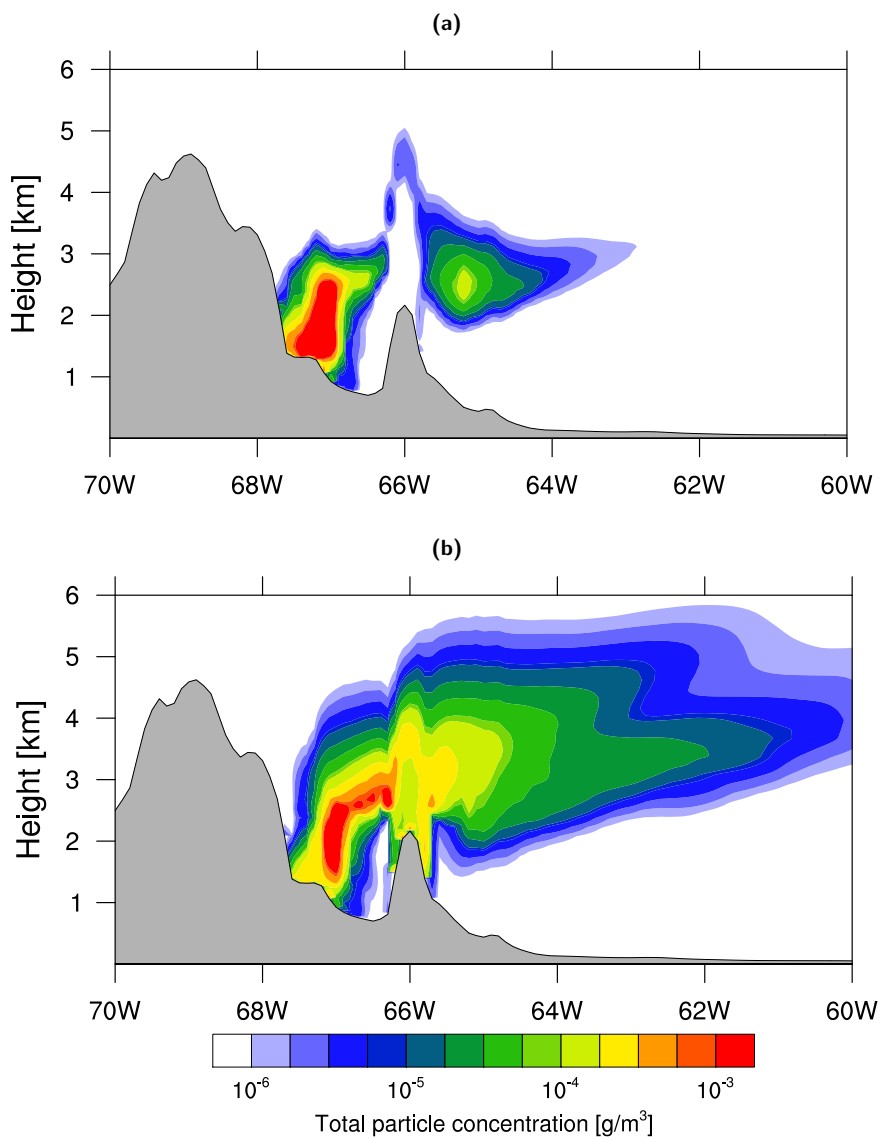

**Figure 16.** Modeled concentration on 13 June at 23:00 UTC. West-East cross section along the red line in Fig. 15 (latitude: 28.2° S). Top panel: FALL3D default configuration. Bottom panel: Alternative configuration.





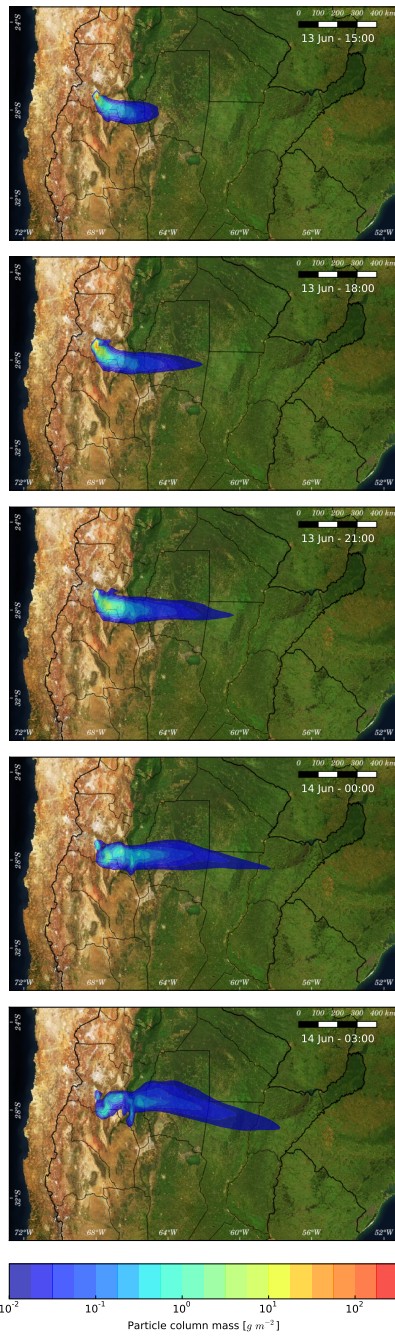

**Figure 17.** Column mass from FALL3D simulations using the alternative configuration. Vertical wind velocity was re-calculated to reduce errors in sloping terrain. Simulation at 15:00, 18:00, 21:00 (13 June), 00:00, and 03:00 UTC (14 June).