# Peer review of "Numerical simulations of windblown dust over complex terrain: The Fiambalá Basin episode in June 2015"

_Atmospheric Chemistry and Physics, 2016_

## Referee Comment (RC1) · L. Mastin (Referee) · 5 Jan 2017

This paper describes simulations of ash resuspension from a 4,400-year-old tephra deposit in the Fiambalá Basin, western Argentina, and examines the accuracy and limitations of current methods to calculate resuspension and transport across complex, mountainous terrain. The study is able to arrive at results close to the observed ones, after using a clever method to modify the vertical wind velocity in their model to account for complex terrain.

I think the paper is well organized, clearly written, the conclusions are justified by the data, and the results are significant in advancing our ability to model ash resuspension. I appreciate the full description of the meteorological situation that produced the

resuspension event. My main criticism is that in section 3.2, which describes the dust emission scheme, the math and some of the physical concepts could be explained more fully. Details are provided below.

I think that the criticisms below can be corrected with minor modification to the paper. I look forward to seeing the final version posted.

Larry Mastin

Specific comments

Page 1, Line 7: change "provided" to "provide"

Page 1, Line 16: change "cloud" to "clouds"

Page 2, line 2: remove "to" after "warned"

Page 2, line 11, add "a" before "product". And change "originated" to "originating"

Page 2, line 13, add "has" after "wind activity"

Page 2, line 14, change "in a major dust source" to "into a major dust source"

Page 4, line 5, change "high elevated" to "high-elevation"

Page 4, line 27: it would be useful to label the southern Puna and northern Pampean Ranges in Figure 1.

Page 5, lines 14-15, "Winds prevailing in middle and upper troposphere favour a dry winter climate over the Puna because the Andes block 15 the westerly flow from the Pacific Ocean". Could you perhaps add a range of latitudes after "over the Puna"? Without knowing what part of the Andes you're talking about, or exactly where the Puna is in Fig. 1, it's hard to know where the winds are favored.

Page 6, line 3. Mention that "PBL" means planetary boundary layer. Also on this line, by "experiments" do you mean model simulations?

Page 6, lines 3-5. I think the concepts mentioned in these lines would benefit from more explanation. What is friction velocity and how does the MYJ scheme provide it for the PBL scheme?

Page 6, lines 7-9. You need a reference for the Noah Land Surface Model, and for the Richard's equation.

Page 6, last line. I thought saltation was bouncing of grains along the ground surface.

Page 7, equation 1. Is alpha dimensionless? Or does it have the same dimensions as $u^2$ (m2/s2)?

Page 7, equation 3. Could you give the units of Q(d)? (kg/m3)*(m3/s3)*(s2/m)=kg/(s*m)? what is its physical meaning? Kilograms per second moved per meter downwind distance?

Page 7, equation 3. Is u* defined? (I can't seem to find a definition). Wind speed at ground level?

Page 7, equation 4. What are the units of p(d) and p(d_s)? Are these mass fractions? Are the data binned?

Page 7, equation 5. I suggest you move the sentence after equation 7 (which explains w) to this point. What are the units of soil moisture? I'm still not sure what "gravimetric soil moisture in percent" is. Mass percent water in the soil?

Page 8, equation 6. What is gamma? An empirical fitting parameter? A physical property?

Page 9, line 26. Change "has propagated" to "had propagated"

Page 10, line 17. What is a two-way nested domain? A domain that is smaller in both x and y?

Page 11, lines 1-7. How did you define the ash source for the Fall3d model? Was the

flux into each source node in the Fall3d model calculated within that model, or was it calculated in the WRF/ARW model and used as input in the Fall3d model?

Page 11, line 33. "In this study, the mass is injected between the two lowest levels". What does this mean? Between the two lowest pressure levels in the WRF/ARW model? Within the second lowest cell in the Fall3d model?

Page 11, Section 5.3. Was the grain-size distribution assumed to be uniform throughout the model domain? Were you assuming that the source material existed only in the Fiambalá Basin?

Page 12, line 3. "the largest friction velocities occurring in steep terrain". How can one identify steep terrain in Figure 8?

Page 12, lines 9-16. It's interesting that u* increases as you increase in model resolution. If you went to still higher resolution, would u* continue to increase? How high must the resolution be in order to simulate a realistic value of u*?

Page 12, lines 20-21. "the scales involved in the mesoscale model are not representative of the spatial scales pertinent to the dust emission processes (i.e., aeolian scales)." I'm not sure what an Aeolian scale is. Can you give an example?

Page 12, line 25. Change "recalculate" to "recalculating"

Page 12, line 27: "Furthermore, the 10-cm depth soil moisture appears to be unrepresentative of the soil surface conditions involved in the dust emission processes." What observations lead you to this conclusion?

Page 13, lines 19-20. Change "an important" to "a significant".

Page 14, line 2. I don't see a definition for x, X, y, Y, or z, Z. Also, in equations 8 and 9, you need to define U, V, W, C, and h.

Page 14, equation 9. This is a clever way to correct for the inaccuracies of low-level wind in a terrain-following coordinate system. Has it been described in any other papers?

Page 14, lines 24-25. "Comparisons among model output 25 and satellite imagery reveal that the modeled dust cloud is further north." Boy, I think the locations looked pretty darned close. However the modeled cloud is wider than the one in the satellite image.

Table 1:

–Terrain and land use. Could you be more specific than "USGS"? What dataset exactly? A footnote with a reference would be helpful.

–Initial and boundary. a footnote with a reference for ERA-Interim would be helpful.

–Minimum and maximum time steps. What does Delta-x mean? You must divide by the maximum wind speed to convert Delta-x (km) to seconds.

–Physics: you need some footnotes or references that explain what the PBL Scheme is (and what the PBL is), Eta Similarity, Eta, the Dudhia scheme, the RRTM scheme etc. Most of the terms under "dynamics" also need footnotes or references.

Figure 1 caption: I suggest mentioning that "OSSB volcanic lineament" refers to the Ojos del Salado-San Buenaventura volcanic lineament. Why do you say at the end of this caption "SRTM, ASTER GDEM is a product of METI and NASA, Imagery GI-Science Research Group @ Heidelberg University." Was this map derived from these data sources?

Figure 2: time on the x axis is local time? Times given in the text are UTC.

Figure 3 caption. You say that the dust clouds are the dashed lines? I don't see dashed lines. But I see pink regions outlined by black, which I assume are the dust clouds.

Figure 4 caption: change "Shaded contour" to "shaded green contour"

Figure 7a. Consider adding the boundaries of the Fall3d model, perhaps in gray outline.

Or maybe add it to Figure 7b if the domain is too small to be seen in Fig. 7a.

Figure 8. Is the outlined region the Fiambalá Basin? Perhaps mention this.

Figure 10. What is the outlined region? Why does the emitted mass abruptly end at the margins of the outlined region?

Figure 11. It's not exactly clear which tick marks the labels refer to on the x axis.

---

## Referee Comment (RC2) · F M Beckett (Referee) · 9 Jan 2017

Summary

Mingari et al. present a study on the meteorological controls on the resuspension of mineral dust from the Fiambala Basin in Argentina. I very much enjoyed reading this paper. It presents an interesting and novel approach to considering the importance of topography when modelling resuspension episodes and I believe represents an important contribution to this field. However, I do have some questions about how the source is set-up in the modelling approach, outlined below, and I would like this to be clarified before publication.

[Figure]

1. Please can you provide further details on how the source area was defined and how the emission was implemented in the modelling?

1.a. How did you define the horizontal grid for the source and is your emission flux independent of this?

On Page 7, line 22 you state that the dust flux is estimated at each point of the deposit. How are these points defined, and at what resolution? My comment relates to your finding that as the horizontal resolution of the met data is increased the total mass emitted increases (Page 12, Line 14). Do you think this is related to a change in the friction velocities in the met data? Or, could it be due to you increasing the number of points from which you can emit particles? As you increase the resolution of your met data grid do you increase the resolution of your source? Does the resolution of the grid from which you can release particles vary or is it independent from the met data grid? Perhaps I just misunderstand how your set-up works here. I wonder if this would be also easier to understand if units for the dust flux presented in Section 3.2.1 were given. Is this in units of mass per metre squared per second?

1.b. Did you identify the whole Fiambala Basin as a possible source area? Please provide details on how this was defined.

1.c. Why did you choose the Shao emission scheme? My understanding was that Folch et al. (2013) showed that using the Westphal scheme best reproduced observations of ash resuspension events during 2011 in Central Patagonia?

2. Do you make a distinction between volcanic ash and dust and do you think you need a different emission scheme when modelling the resuspension of mineral dust as opposed to fresh volcanic ash deposits? Can this be accounted for in your threshold friction velocity? In relation to this it might then be worth mentioning in the Introduction that resuspension events of volcanic ash occur in other areas of the world, including Alaska and Iceland. Indeed the London VAAC provide daily resuspended ash forecasts to the Icelandic Met Office (see Leadbetter et al. 2012, JGR, doi:10.1029/2011JD016802).

3. Please clarify what particle size range you released. On page 7 Line 25 you state that the source term was calculated for particles with diameter less than 60 microns. However, on Page 11 you state that you used a log normal distribution with particles ranging in diameter from 0 phi to 8 phi (1 mm – 4 microns). Also, you suggest that only particles with diameter less than 60 microns can be suspended, but there have been observations of much larger volcanic ash grains being resuspended in Iceland (up to 177 microns, see Liu et al., 2014, Ash mists and brown snow: Remobilization of volcanic ash from recent Icelandic eruptions, J. Geophys. Res. Atmos., 119, doi:10.1002/2014JD021598).

4. Please also clarify how the particles were released. On page 12, Line 3, you state that the mass was injected between 10 and 160 m a.gl. Was this as a uniform distribution? Also, how did you decide on the top height of 160 m a.g.l?

5. Please comment as to whether your approach to represent soil moisture in your calculation of the threshold friction velocity accounts for precipitation and wetting and drying of the deposit. You note that there is an offset in the observed ash in the satellite retrievals and the peak modelled mass loadings. You suggest that this could be related to the height of the ash in the atmosphere, with the peak height occurring later in the event. But, I would interpret Figure 14 as showing that the satellite detection period continued into Sunday 14th June 2015 when the modelled ash cloud height was very low. Could the offset be related to the time needed to wet and dry deposits as noted in the study by Leadbetter et al. (2012), or is your application of the threshold friction velocity capturing this behaviour?

6. Finally, please can you provide more detailed Figure captions, so that the reader can interpret the figures without having to refer back to the text. The caption for Figure 11 states that the results are very sensitive to the horizontal resolution of the WRF met fields; it would be nice to expand on this. The caption for Figure 2 states that the signatures of a zonda episode are observed, again please expand on this. And, Figure 16, what is the 'alternative configuration'?

Minor Comments

In the abstract you refer to 'further studies' which concluded that the observed ash cloud was from a resuspension event. This comes across as an unsubstantiated comment. Perhaps you could re-word, or introduce the Collini et al. (2015) reference.

Page 1, line 12, indicated should be indicate.

Page 1, Line 16, addition of 'the' such that the sentence reads: '.....were also performed to model THE dust cloud spreading.....'

Page 2, Line 24, please re-word. For example: 'These eruptions blanketed a vast area of Patagonia in Argentina with volcanic ash'.

Page 2, Line 33, replace 'particularly'. For example: 'However, studies focussed on ......'

Page 4, Line 17, re-word sentence. For example: 'The Cerro Blanco Volcanic Complex (CBVC) is located ON the eastern edge of the Ojos del Salado-San Buenaventura volcanic lineament (Fig. 1) and IS defined as a nested calderas system with ASSOCIATED domes and pyroclastic deposits'

Page 4, Line 32, spelling correction: 'Wind activity continued to mobilize this pyroclastic material until today turning the Fiambalá Basin INTO one of the major dust sources in North-Western Argentina'

Page 5, Line 26. What does ARW stand for when referring to the core of the WRF model?

Page 6, Line 19. You state that you use the Ganser scheme to calculate the particle settling velocity. Do you assume the particles are spherical or assign a shape parameter?

Page 7. Please clarify the introduction to the vertical flux scheme. I think the initial sentence needs to be re-worded. What experiment are you referring to when you state

that: 'The experiment confirms that the saltation ....'

Page 8, Line 3. Please re-word this sentence, for example: 'Note that the effect of the roughness elements on the threshold friction velocity is not considered'.

Page 8, Line 15. Please re-word this sentence, for example: 'Dusty winds with gusts OF up to 90 km/h CAUSED A RANGE OF PROBLEMS, such as trees..'

Page 9, Line 2. I am not sure I understand what is meant by 'a strong drying'. Please clarify.

Page 9, Line 11. Please correct spelling and re-word, for example: 'The dust product is AN RGB (Red-Green-Blue) composite based upon infrared channels, WHICH ALLOWS suspended dust TO BE DETECTED during both THE day and night TIME.

Page 9, Line 27. Should read: 'Green shaded AREA'.

Figure 4. What are the purple/blue areas that are shaded?

Page 11. What are theta-e contours?

Page 12, Line 27. Spelling Mesoscale.

Frances Beckett
* * *

---

## Author Comment (AC1) · 28 Mar 2017

The comment was uploaded in the form of a supplement:
http://www.atmos-chem-phys-discuss.net/acp-2016-851/acp-2016-851-AC1-supplement.pdf

---

## Author Response (AR1)

**Response to reviews of acp-2016-851**

We thank the reviewers for their constructive comments and careful evaluations. Our responses to comments are provided below (reviews' comments in italics and additions and removals to the manuscript in blue and red text, respectively).

In the revised version of the paper, much attention has been focused on improving section 3.2.1: "Dust emission scheme" and section 5.3: "Source strength". Following the reviewers' suggestion, we provided more details on the dust emission scheme and its implementation.

**Referee #1: L. Mastin**

Page 4, line 27: it would be useful to label the southern Puna and northern Pampean Ranges in Figure 1.

Page 5, lines 14-15, "Winds prevailing in middle and upper troposphere favour a dry winter climate over the Puna because the Andes block 15 the westerly flow from the Pacific Ocean". Could you perhaps add a range of latitudes after "over the Puna"? Without knowing what part of the Andes you're talking about, or exactly where the Puna is in Fig. 1, it's hard to know where the winds are favored.

Figure 1 caption: I suggest mentioning that "OSSB volcanic lineament" refers to the Ojos del Salado-San Buenaventura volcanic lineament. Why do you say at the end of this caption "SRTM, ASTER GDEM is a product of METI and NASA, Imagery GI-Science Research Group @ Heidelberg University." Was this map derived from these data sources?

Response: The range of latitudes of the Puna plateau is indicated at the beginning of the paragraph (section 2.2, p. 5, l. 10). Figure 1 was modified as suggested by referee and a new background map is used (see section: Modifications in figures).

Changes in section 2.2 (p. 5, l. 10): In the Puna Andean plateau (22°-28°28°-22° S, northwestern Argentina), dry salt basins and lake beds are important dust source...

Page 6, lines 3-5. I think the concepts mentioned in these lines would benefit from more explanation. What is friction velocity and how does the MYJ scheme provide it for the PBL scheme?

Page 7, equation 3. Is u\* defined? (I can't seem to find a definition). Wind speed at ground level?

Response: More details are provided in section 3.1, including a friction velocity definition.

The PBL and surface layer schemes are composed of physics modules working together within the WRF-ARW model (internally). The surface scheme provide a lower boundary condition for the PBL scheme.

Changes in section 3.1 (p. 6, l. 3): The experiments are conducted with a local closure PBL scheme implemented by Mellor-Yamada-Janjić (MYJ). The MYJ scheme runs in concurrence with The friction velocity, u\*, is an important scaling variable in the similarity theory (Monin and Obukhov, 1954; Janjic, 1996) related to wind shear near the surface and is defined as

$$u_* = \sqrt{\frac{|\tau|}{\rho}} \tag{1}$$

where  $\tau$  is the surface Reynold's stress and  $\rho$  the density (Stull, 1988). The friction velocity is calculated by the Eta surface layer scheme, which is based on the similarity theory, and provides the friction velocity,  $u_*$ , for the PBL scheme.

 Changes in section 3.1 (p. 6, l. 16): A satisfactory description of the transport of aerosols depends on an accurate representation of the planetary boundary layer (PBL) (Banks et al., 2016). The numerical experiments are conducted with a local closure PBL scheme implemented by Mellor-Yamada-Janjić (MYJ) (Janjic, 1994). The surface schemes calculate the surface fluxes of momentum, moisture, and heat and provide a lower boundary condition for the PBL scheme (Skamarock et al., 2008).

Page 7, equation 1. Is alpha dimensionless? Or does it have the same dimensions as  $u^2 (m2/s^2)$ ?

Page 8, equation 6. What is gamma? An empirical fitting parameter? A physical property?

Page 7, equation 3. Could you give the units of Q(d)? (kg/m3)\*(m3/s3)\*(s2/m)=kg/(s\*m)? what is its physical meaning? Kilograms per second moved per meter downwind distance?

Page 6, last line. I thought saltation was bouncing of grains along the ground surface.

Page 7, equation 4. What are the units of p(d) and  $p(d_s)$ ? Are these mass fractions? Are the data binned?

Page 7, equation 5. I suggest you move the sentence after equation 7 (which explains w) to this point. What are the units of soil moisture? I'm still not sure what "gravimetric soil moisture in percent" is. Mass percent water in the soil?

Response: We re-write the section 3.2 in order to more fully explain the dust emission scheme indicating units of relevant magnitudes.

Specifically,  $\alpha$  has the units  $m s^{-2}$  and  $\gamma$  is an experimental parameter in  $kg s^{-2}$  taking into account the effect of inter-particle cohesion. The vertically-integrated streamwise saltation flux Q ( $kg m^{-1} s^{-1}$ ) is the vertical integral of the streamwise saltation flux, q(z) ( $kg m^{-2} s^{-1}$ ):

$$Q = \int_0^\infty q(z)\,dz$$

We provide a clearer definition of the saltation mode of movement.

The particle grain size distribution is discretized in a series of bins, with p(d) representing the mass fraction of the class characterized by particles of size d.

Finally, the gravimetric soil moisture is the ratio of mass of water to dry mass of soil sample.

- Changes in section 3.2.1 (p. 7, l. 13): Saltation is the most common form of aeolian sand transport. It refers to the hopping motion of sand particles (tipically, d ~ 70 500 μmfollow a smooth trajectory path, mostly unaffected by turbulence. Such trajectories define a motion called *saltation*) along the ground surface in the direction of the wind (Shao, 2008).
- Changes in section 3.2.1 (p. 7, l. 32): The wind tunnel experiments conducted by indicate that the vertical flux of vertically-integrated streamwise saltation flux Q ( $kgm^{-1}s^{-1}$ ), is calculated in this paper using the expression suggested by Owen (1964) for the case of saltating particles of size d is proportional to  $Q(d_s)$ , the horizontal flux of  $d_s$ :
- Changes in section 3.2.1 (p. 8, l. 15): We assume a grain size distribution discretized in a group of particle classes. A class of particles of size d is characterized by a mass fraction p(d) satisfying  $\sum p(d) = 1$ .
- Changes in section 3.2.1 (p. 9, l. 13): where f(w) is a moisture correction and w is the gravimetric soil moisture, i.e. the water content of a soil sample expressed as a percent of the oven-dry mass of the sample.

Page 11, lines 1-7. How did you define the ash source for the Fall3d model? Was the flux into each source node in the Fall3d model calculated within that model, or was it calculated in the WRF/ARW model and used as input in the Fall3d model?

Response: The dust flux is calculated by the FALL3D dispersal model using input data from the  $\overline{\rm WRF/ARW}$  model.

• Changes in section 5.3 (p. 13, l. 9): The emission rates are obtained as a pre-processing step by the FALL3D dispersal model using the emission scheme proposed by Shao and Leslie (1997) (thereinafter "Shao scheme"). The vertical flux of dust, Eq. (5), is calculated on the FALL3D fine domain with 0.01° grid resolution (see section 5.1). Since the WRF-ARW fields were interpolated onto the same FALL3D grid, results obtained using meteorological data at different resolutions can be compared.

Page 11, line 33. "In this study, the mass is injected between the two lowest levels". What does this mean? Between the two lowest pressure levels in the WRF/ARW model? Within the second lowest cell in the Fall3d model?

Response: Actually, we should say: "the mass is injected between the two lowest layers" (of the FALL3D model). We corrected this statement and provided more details about the injection height in dust models.

• Changes in section 5.3 (p. 13, l. 18): The elevation at which dust aerosols are injected into the atmosphere has a strong influence on the subsequent transport. The initial height for the emitted dust (injection height) is highly uncertain in numerical models (Darmenov, 2009). For example, Park et al. (2007) considered two injection schemes depending on whether the mass is injected in the first model layer (1-layer injection) or in the lowest three layers (3-layer injection). The author suggested that the 1-layer injection scheme can significantly underestimate the amount of dust transported. Conversely, the injection height in the CFORS regional model (Uno et al., 2003) is determined from the vertical profile of the potential temperature. In this study, the time-dependent source term is obtained assuming a uniform vertical distribution of the emitted mass up to a fixed injection height. We assume that the mass is injected between the two lowest layers (top layer height: 260 m a.g.l.) of the FALL3D model.

Page 11, Section 5.3. Was the grain-size distribution assumed to be uniform throughout the model domain? Were you assuming that the source material existed only in the Fiambalá Basin?

Response: Yes, you're right. More details are provided now.

• Changes in section 5.3 (p. 13, l. 13): We have no accurate information on the distribution of deposits in this region. Consequently, we consider that the emission area is delimited by the solid line in Fig. 7, defined approximately by the terrain height contour of 2000 meters. According to the Shao scheme, Eq. (5) This area represents properly the origin of the dust cloud detected by satellite imagery (Fig. 3). In our modeling strategy, dust emission is assumed to take place within the spatial region determined by the solid line, with no emission occurring outside (see Fig. 10).

Page 12, line 3. "the largest friction velocities occurring in steep terrain". How can one identify steep terrain in Figure 8?

Response: Figure 8 must be compared with Fig. 7.

• Changes in section 5.3 (p. 14, l. 5): Figure 8 shows that the friction velocity for the innermost WRF domain (2 km) is strongly correlated with the orographic features of the region, with the largest friction velocities occurring in steep terrain (compare Fig. 8 with Fig. 7).

Page 12, lines 9-16. It's interesting that  $u^*$  increases as you increase in model resolution. If you went to still higher resolution, would  $u^*$  continue to increase? How high must the resolution be in order to simulate a realistic value of  $u^*$ ?

Response: This is a difficult matter, which is not resolved in this work. However, a comparison between  $\overline{\text{Fig. 2}}$  (ground measurements) and Fig. 9c (numerical simulations) gives a clue that the results with a model resolution of 2 km are reasonable (at least in the cases of 2-m temperature and dew point temperature).

Page 12, lines 20-21. "the scales involved in the mesoscale model are not representative of the spatial scales pertinent to the dust emission processes (i.e., aeolian scales)." I'm not sure what an Aeolian scale is. Can you give an example?

Response: Below are provided typical values of roughness lengths for erodible surfaces.

Changes in section 5.3 (p. 14, l. 26): As noted Darmenova et al. (2009), roughness lengths relevant to the aeolian processes, with typical values for erodible surfaces ranging from 10-3 cm to 1 cm, may differ significantly from the mesoscale model roughness lengthsoperate at seales unsuitable for dust emission parametrizations.

Page 12, line 27: "Furthermore, the 10-cm depth soil moisture appears to be unrepresentative of the soil surface conditions involved in the dust emission processes." What observations lead you to this conclusion?

Response: According to Shao (2008): "as far as wind erosion is concerned, we are mostly interested in the soil moisture of the very top layer, which is probably less than 10 mm deep." (Shao, Y.: Physics and modelling of wind erosion, chapter 4.2, 2 edn., 2008.).

Reported values of 10-cm depth soil moistures can be an order of magnitude larger than values in the top 0-1 cm soil layer (see Darmenova et al.: Development of a physically based dust emission module within the Weather Research and Forecasting (WRF) model: Assessment of dust emission parameterizations and input parameters for source regions in Central and East Asia, JGR., 114, 2009).

On the other hand, we have recently observed a significantly better level of forecast skill for resuspension of fresh volcanic deposits in Patagonia after precipitation periods using 1-cm depth soil moistures (unpublished results).

Page 14, equation 9. This is a clever way to correct for the inaccuracies of low-level wind in a terrain-following coordinate system. Has it been described in any other papers?

Response: This correction is appropriate for the simple terrain-following coordinate system used by  $\overline{\text{FALL3D}}$  model. We have no knowledge of the existence of another dispersion model with such a coordinate system.

**Table 1:**

-Terrain and land use. Could you be more specific than "USGS"? What dataset exactly? A footnote with a reference would be helpful.

-Initial and boundary. a footnote with a reference for ERA-Interim would be helpful.

-Minimum and maximum time steps. What does Delta-x mean? You must divide by the maximum wind speed to convert Delta-x (km) to seconds.

-Physics: you need some footnotes or references that explain what the PBL Scheme is (and what the PBL is), Eta Similarity, Eta, the Dudhia scheme, the RRTM scheme etc. Most of the terms under "dynamics" also need footnotes or references.

Response: We added footnotes and a reference.

We set the minimum and maximum time steps for each domain from the horizontal resolution:  $\Delta x$ . For example, the minimum timestep for the coarsest domain ( $\Delta x = 18 \, km$ ) is 18 seconds ( $18 \times 1$ ) and the maximum timestep is 108 seconds ( $18 \times 6$ ). This criterion assures us numerical stability.

• Changes in table 1 (footnotes): aStatic data from the U.S. Geological Survey (USGS): 24-category USGS-based land use and topography data from the USGS 30 arc second resolution global dataset (GTOPO30). bERA-Interim re-analysis data (model levels and surface) obtained from the ECMWF

(European Center for Medium-range Weather Forecasts).  ${}^{c}\Delta x$ : horizontal spatial resolution of the grid domain.  ${}^{d}$ Planetary Boundary Layer scheme (see Skamarock et al., 2008, for further details).

Figure 2: time on the x axis is local time? Times given in the text are UTC. Response: Times are given in UTC in this work.

Figure 10. What is the outlined region? Why does the emitted mass abruptly end at the margins of the outlined region?

Response: The outlined region is the emission area, defined approximately by the terrain height contour  $\overline{\text{of }2000}$  meters. This area represents properly the origin of the dust cloud detected by satellite imagery. It is assumed that dust emission to take place within this spatial region, with no emission occurring outside.

Figure 11. It's not exactly clear which tick marks the labels refer to on the x axis.

Response: We removed minor ticks in Fig. 11 (see section: Modifications in figures).

**Referee #2: F. Beckett**

1. Please can you provide further details on how the source area was defined and how the emission was implemented in the modelling?

1.a. How did you define the horizontal grid for the source and is your emission flux independent of this? On Page 7, line 22 you state that the dust flux is estimated at each point of the deposit. How are these points defined, and at what resolution? My comment relates to your finding that as the horizontal resolution of the met data is increased the total mass emitted increases (Page 12, Line 14). Do you think this is related to a change in the friction velocities in the met data? Or, could it be due to you increasing the number of points from which you can emit particles? As you increase the resolution of your met data grid do you increase the resolution of your source? Does the resolution of the grid from which you can release particles vary or is it independent from the met data grid? Perhaps I just misunderstand how your set-up works here. I wonder if this would be also easier to understand if units for the dust flux presented in Section 3.2.1 were given. Is this in units of mass per metre squared per second?

1.b. Did you identify the whole Fiambala Basin as a possible source area? Please provide details on how this was defined.

1.c. Why did you choose the Shao emission scheme? My understanding was that Folch et al. (2013) showed that using the Westphal scheme best reproduced observations of ash resuspension events during 2011 in Central Patagonia?

Response: More details of the implementation of the emission source are given in section 5.3.

1.a. The meteorological fields were interpolated onto the same FALL3D grid (with the same number of potential emission points) and differences in the emitted mass is a direct consequence of changes in the friction velocities.

We re-write the sections 3.2 in order to more fully explain the dust emission scheme, including units of relevant magnitudes. Specifically, the dust emission flux has the units:  $kg m^{-2} s^{-1}$ .

• Changes in section 5.3 (p. 13, l. 10): The vertical flux of dust, Eq. (5), is calculated on the FALL3D fine domain with 0.01° grid resolution (see section 5.1). Since the WRF-ARW fields were interpolated onto the same FALL3D grid, results obtained using meteorological data at different resolutions can be compared.

1.b. We identify the whole Fiambala Basin as a possible source area, with no emission occurring outside. This is an assumption, but it is consistent with satellite imagery.

• Changes in section 5.3 (p. 13, l. 13): We have no accurate information on the distribution of deposits in this region. Consequently, we consider that the emission area is delimited by the solid line in Fig. 7, defined approximately by the terrain height contour of 2000 meters. According to the Shao scheme, Eq. (3) This area represents properly the origin of the dust cloud detected by satellite imagery (Fig. 3). In our modeling strategy, dust emission is assumed to take place within the spatial region determined by the solid line, with no emission occurring outside (see Fig. 10).

1.c. There are two reasons: firstly, the Shao scheme is the most realistic scheme of those implemented in the FALL3D model. However, it is strongly dependent on the grain size distribution. The poor performance observed by Folch et al. (2014), is possibly related to the uncertainty of the size distribution (indeed, a modeled distribution with large uncertainties was used in that work). Fortunately, in this work we have a grain size distribution based on field measurements. Secondly, we used an updated implementation of the Shao emission scheme with some modifications (compare Eq. (5) with Eq. (11) in Folch et al. (2014)).

2. Do you make a distinction between volcanic ash and dust and do you think you need a different emission scheme when modelling the resuspension of mineral dust as opposed to fresh volcanic ash deposits? Can this be accounted for in your threshold friction velocity? In relation to this it might then be worth mentioning in the Introduction that resuspension events of volcanic ash occur in other areas of the world, including Alaska and Iceland. Indeed the London VAAC provide daily resuspended ash forecasts to the Icelandic Met Office (see Leadbetter et al. 2012, JGR, doi:10.1029/2011JD016802). Response: We distinguish in this work between mineral dust and volcanic ash. We follow the criterion defined in "Volcanic Ash versus Mineral Dust: Atmospheric Processing and Environmental and Climate Impacts", B. Langmann. ISRN Atmospheric Sciences, vol. 2013, 2013 (doi:10.1155/2013/245076):

"Weathered mineral dust may originate from volcanic tephra (tephra is defined as any fragmental material produced by a volcanic eruption regardless of composition and fragment size); however, volcanic ash represents relatively fresh material produced during a recent volcanic eruption (recent in this context means no longer than about 100 years ago) and is therefore different form mineral dust."

For this reason, we consider more reasonable the choice of an emission scheme for mineral dust in this case.

On the other hand, we think it is necessary to develop a specific scheme for fresh ash deposits. However, the Shao scheme may be adequate with slight modifications. In fact, this scheme is based on theoretical results of general validity, but some experimental parameters, such as  $\alpha$  (associated to the efficiency of saltation bombardment) and  $\gamma$  (associated to the inter-particle cohesion), should be determined in the case of fresh ash deposits.

We thank you for the information on ash resuspension events in other parts of the world. We added this information in the introduction.

• Changes in the introduction (p. 2, l. 32): Additionally, remarkable events of volcanic ash resuspension occur in other areas of the world, including Alaska (Hadley et. al. 2004) and Iceland (Leadbetter et al., 2012; Liu et al., 2014). In fact, the London VAAC provides daily resuspended ash forecasts to the Icelandic Met Office.

3. Please clarify what particle size range you released. On page 7 Line 25 you state that the source term was calculated for particles with diameter less than 60 microns. However, on Page 11 you state that you used a log normal distribution with particles ranging in diameter from 0 phi to 8 phi (1 mm – 4 microns). Also, you suggest that only particles with diameter less than 60 microns can be suspended, but there have been observations of much larger volcanic ash grains being resuspended in Iceland (up to 177 microns, see Liu et al., 2014, Ash mists and brown snow: Remobilization of volcanic ash from recent Icelandic eruptions, J. Geophys. Res. Atmos., 119, doi:10.1002/2014JD021598).

Response: The dust flux of fine particles depends on the fraction mass of coarse particles through the saltation flux, Eq.(2) and (5). As a consequence, the full size distribution is used to calculated the flux of the finest particles.

We remove the statement: "only particles with diameter less than 60 microns can be suspended", but it is indicated that the typical size for suspended particles is  $d \leq 70 \,\mu m$  (Shao, 2008). References to Liu et al. (2014) and Uno et al., (2006) are added to justify the election of  $60 \,\mu m$  as the maximum particle size. Indeed, Liu et al., (2014) showed that most of the mass was contained within the  $32 - 63 \,\mu m$  size fraction. Consequently, the limit value  $d = 60 \,\mu m$  seems to be a reasonable choice.

- Changes in section 3.2.1 (p. 7, l. 9): Suspension refers to the floating motion of airborne fine particles (tipically, d ≤ 70 µm) (Shao, 2008).
- Changes in section 3.2.1 (p. 8, l. 19): The Most of dust models consider a dust size range with maximum diameter between  $40 \,\mu m$  and  $76 \,\mu m$  (Uno et al., 2006). However, larger particles can be transported by wind over significant distances. As an example, measurements of particle size distribution for volcanic ash in Iceland showed that ash particles as large as  $177 \,\mu m$  can be remobilized over several hundred kilometers (Liu et al., 2014). The authors mention that most of the mass was contained within the  $32 63 \,\mu m$  size fraction. In this paper, the source term for the dispersion model was calculated using Eq. (5) for particle sizes  $d < 60 \,\mu m$  in order to take into account only the suspended particles.
- Changes in section 5.3 (p. 13, l. 26): According to the Shao scheme, the emission rate depends on the particle grain size distribution p(d) of the deposit. We used a log-normal distribution We assume a grain size distribution uniform throughout the model domain discretized along 14 classes. Each particle class is characterized by its particle size d, density  $\rho$ , and shape. We used a log-normal distribution ranging from  $\phi = 0$  to  $\phi = 8$  ( $\phi = -\log_2(d)$ , with d in mm). The parameters of

the distribution, obtained from field measurements, are: mean,  $\overline{\phi} = 3\phi$ , and standard deviation,  $\sigma_{\phi} = 1,5\phi$  (J. G. Viramonte, personal communication, 2015). The density of particles varies linearly with  $\phi$  between 1.2-2.2 g cm-3.

The elevation at which dust aerosols and a constant sphericity  $\Phi = 0.9$  is assumed. Only the finest particles  $(d < 60 \,\mu m)$  are injected into the atmospherehas a strong influence on the subsequent transport. The time-dependent source term is obtained assuming an uniform vertical distribution of the emitted mass up to a fixed injection height. In this study, the mass is injected between the two lowest levels (10 a.g.l. and 160 a.g.l.) (see section 3.2.1 for further details). However, the saltation flow, Q, must be estimated for each particle class since the full distribution ( $\phi = 0 - 8$ ) is required in Eq. (5) to calculate the vertical flux of the finest particles.

4. Please also clarify how the particles were released. On page 12, Line 3, you state that the mass was injected between 10 and 160 m a.gl. Was this as a uniform distribution? Also, how did you decide on the top height of 160 m a.g.l?

Response: The initial injection height is highly uncertain in dust models. We follow a criterion intermediate to that of Park et al. (2007), i.e., the emitted mass is uniformly distributed between the two lowest model layers (centered at 10 and 160 m a.gl., respectively, with top height at 235 m a.g.l. in our case). However, a more rigorous approach is advisable.

• Changes in section 5.3 (p. 13, l. 18): The elevation at which dust aerosols are injected into the atmosphere has a strong influence on the subsequent transport. The initial height for the emitted dust (injection height) is highly uncertain in numerical models (Darmenov, 2009). For example, Park et al. (2007) considered two injection schemes depending on whether the mass is injected in the first model layer (1-layer injection) or in the lowest three layers (3-layer injection). The author suggested that the 1-layer injection scheme can significantly underestimate the amount of dust transported. Conversely, the injection height in the CFORS regional model (Uno et al., 2003) is determined from the vertical profile of the potential temperature. In this study, the time-dependent source term is obtained assuming a uniform vertical distribution of the emitted mass up to a fixed injection height. We assume that the mass is injected between the two lowest layers (top height: 235 m a.g.l.) of the FALL3D model.

5. Please comment as to whether your approach to represent soil moisture in your calculation of the threshold friction velocity accounts for precipitation and wetting and drying of the deposit. You note that there is an offset in the observed ash in the satellite retrievals and the peak modelled mass loadings. You suggest that this could be related to the height of the ash in the atmosphere, with the peak height occurring later in the event. But, I would interpret Figure 14 as showing that the satellite detection period continued into Sunday 14th June 2015 when the modelled ash cloud height was very low. Could the offset be related to the time needed to wet and dry deposits as noted in the study by Leadbetter et al. (2012), or is your application of the threshold friction velocity capturing this behaviour?

Response: The ratio of wet and dry threshold friction velocity is taken into account through the empirical expression obtained by Fécan, Eq. (8), using the soil moisture calculated within the WRF-ARW model by the Noah land surface model.

However, the 10-cm depth soil moistures used in this work appears to be unrepresentative of the soil surface conditions. In fact, these values can be an order of magnitude larger than the top 0-1 cm soil layer, more relevant for dust emission (see Darmenova et al.: Development of a physically based dust emission module within the Weather Research and Forecasting (WRF) model: Assessment of dust emission parameterizations and input parameters for source regions in Central and East Asia, JGR., 114, 2009).

As a consequence, the problem found by Leadbetter et al. (2012): "Analysis shows that all of the false predictions of episodes of resuspended ash in late January and early February 2011 occur after periods of precipitation" is a problem found by us in other case studies after periods of precipitation. In fact, we have recently observed a significantly better level of forecast skill for resuspension of fresh volcanic deposits in Patagonia after precipitation periods using the modeled 1-cm depth soil moistures instead

of 10-cm depth soil moistures (unpublished results). However, we think that the offset in the satellite retrievals and the modelled emission rate in the present work is not associated to this problem due to the absence of precipitation during the previous days (indeed, austral winter is a dry season in this region).

Finally, we regret that the figure 14 was not explained clearly. This figure represents the cloud height over the basin (i.e. the altitude of the freshly emitted dust). But the satellite detection period continued into 14th June ONLY far from the basin. To avoid this confusion, only the beginning of the satellite detection period is indicated in Fig. 14 and a more detailed caption is provided (see section: Modifications in figures).

• Changes in section 5.4 (p. 15, l. 21): In order to quantify the dust injection height, we define the cloud top height at a specific location as the maximum height at which a concentration of  $1 mg m^{-3}$  is attained (i.e. the highest contour in Fig. 12). Figure 14 shows the time series for the cloud top height over the Fiambalá Basin, i.e. the cloud height averaged over the FALL3D domainfine domain (28–27° S and 68–67.3° W).

6. Finally, please can you provide more detailed Figure captions, so that the reader can interpret the figures without having to refer back to the text. The caption for Figure 11 states that the results are very sensitive to the horizontal resolution of the WRF met fields; it would be nice to expand on this. The caption for Figure 2 states that the signatures of a zonda episode are observed, again please expand on this. And, Figure 16, what is the 'alternative configuration'?

Response: Thank you very much for your suggestion. We provide more detailed captions now.

- Changes in Fig. 2 caption: Measurements from an automatic weather station from the National Weather Service of Argentina at Catamarca Airport, 200 km from the Fiambalá Basin. Wind speed (red line with circle markers), temperature (T, black line), and dew point temperature ( $T_D$ , blue line). The signatures of a Zonda episode, i.e. warm, dry and windy conditions, are observed on 13 June.
- Changes in Fig. 11 caption: Time series for the emission rates rate (spatially integrated) according to Shao emission schemedriven by meteorological inputs at different horizontal resolutions (2 km, 6 km, and 18 km). Results are very The dust flux is remarkable sensitive to horizontal friction velocity resolution. The gray-shaded area represents the time window with satellite detection of WRF meteorological fields airborne dust. According to simulations, the emission occurred before satellite detection.
- Changes in Fig. 16 caption: Modeled concentration on 13 June at 23:00 UTC. West-East cross section along the red line in Fig. 15 (latitude: 28.2° S). Two numerical schemes are tested. Top panel: FALL3D default configuration. Bottom panel: Alternative configuration using a numerical scheme to more accurately compute the horizontal advection in abrupt terrains (see Eq. (10)).

Page 6, Line 19. You state that you use the Ganser scheme to calculate the particle settling velocity. Do you assume the particles are spherical or assign a shape parameter?

Response: A constant sphericity  $\Phi = 0.9$  is assumed.

Page 9, Line 2. I am not sure I understand what is meant by 'a strong drying'. Please clarify.

Response: We clarified this sentence.

• Changes in section 4 (p. 10, l. 12): The automatic weather station measured since 03:00 UTC a strong drying reduction in relative humidity (sudden decrease in  $T_D$  and increase in T) until 20:00 UTC, and a strong increase in wind speed, with a maximum speed of ...

Figure 4. What are the purple/blue areas that are shaded?

Response: We forgot to specify this information. We are sorry.

• Changes in Fig. 4 (caption): Meteorological situation corresponding to 12:00 UTC on 12 June 2015 (a) and 12:00 UTC on 13 June 2015 (b). Solid line: the 500 hPa geopotential height (gpm). Shaded green contour: Wind speed >  $24 m s^{-1}$ . Shaded blue contour: Wind speed >  $48 m s^{-1}$ . Fiambalá location is stated by a red circle. Dataset from ERA-Interim re-analysis.

Page 11. What are theta-e contours?

Response: It refers to equivalent potential temperature.

• Changes in section 5.2 (p. 12, l. 30): Figure 9a shows a vertical cross section of equivalent potential temperature (or theta-e) and vertical wind...

**Modifications in figures**